# Multistage Economic Scheduling Model of Micro-Energy Grids Considering Flexible Capacity Allocation

Hang Liu [1,*], Yongcheng Wang [1], Shilin Nie [2], Yi Wang [1] and Yu Chen [1]

1 School of Management Engineering, Zhengzhou University of Aeronautics, Zhengzhou 450046, China; wyc@zua.edu.cn (Y.W.); wangyi413@outlook.com (Y.W.); chenyu318233047@outlook.com (Y.C.)
2 Zhangshu Development and Reform Commission, Zhangshu 331200, China; nsl03ncepu@163.com
* Correspondence: liuhang@zua.edu.cn

**Abstract:** Micro-energy grids integrating multiple energy sources can realize the efficient use of renewable energy and accelerate the process of energy transition. However, due to the uncertainty of renewable energy, the stability and security of system operations should be taken into account with respect to multi-energy coupling economic operations. Thus, it is essential to make flexible capacity allocations in advance of the actual scheduling of production in the micro-energy grid. With this motivation, this paper constructs a three-stage scheduling model corresponding to the running stage of the spot market. Specifically, the capacity of flexible, active devices is configured in the day-ahead stage; then, the intraday economic operation dispatching scheme is provided according to the capacity configuration. Based on the day-ahead and intraday optimization results, the system power balance is realized through the dispatching process using the reserve capacity of flexible active devices for deviations generated in the real-time stage of renewable energy. For the uncertainty of renewable energy output, the clustering method is applied to realize the clustering analysis of renewable energy output scenarios. In addition, the conditional value at risk (CVaR) theory is introduced to modify the three-stage stochastic optimization model, and the risk values caused by uncertainty are quantitatively evaluated. Finally, we simulate a practical case to verify the effectiveness of the proposed model. The results show that day-ahead flexible capacity allocation enhances the autonomy of the micro-energy grid system, ensures a certain degree of system operational security, and reduces balancing costs in the real-time stage. The higher the risk aversion factor, the more operational costs the system operator pays to avoid the risk. In addition, if the carbon penalty coefficient is higher, the overall carbon emission level of the micro-energy grid will decrease, but it will gradually converge to a minimal level. This paper guides the development of micro-energy grids and has important constructional significance for the construction of multi-energy collaborative mechanisms.

**Keywords:** micro-energy grids; multistage optimization; capacity allocation; CVaR; processing of uncertainty

## 1. Introduction

With the continuous growth of the social economy and the increasing energy demand, the energy supply industry faces the dual pressure of meeting demand and reducing carbon emissions. Therefore, the utilization of renewable energy has attracted more and more attention and favor. By using locally distributed renewable energy, the micro-energy grid not only has a high proportion of renewable energy access but also uses various energy conversion and energy storage devices to achieve the comprehensive utilization of various forms of energy, which effectively improves energy utilization efficiency, reduces pollutant emissions, and has strong economic and environmental protection. It is known as one of the main trends in the development of emerging energy systems, and it also provides an important new model of resource utilization for the energy industry.

## 1.1. Application Scenarios of Micro-Energy Grid

Micro-energy grids have extensive application scenarios in industrial parks, large office buildings, and residential communities. For example, Reference [1] used an urban community as the application scenario to study the optimization model of micro-energy grids with compressed air energy storage devices. The operation results verify the validity of the model. Reference [2] took the energy supply of a hotel as an example. It mainly discussed the optimization of combined cooling, heating, and power (CCHP) unit capacity and the power-cooling ratio under uncertain demand (cold, heat, and electricity), solar radiation, and prices. Reference [3] took the energy network of a residential community as an example, using the geothermal heat and waste heat generated by the cement industry combined with CCHP units to meet the cooling, heating, and electricity needs of the community and to conduct an economic analysis; the results show that the proposed distributed energy has economic viability for the system. Reference [4] took a rural microgrid as the research object, and comprehensively considered the seasonal characteristics of the rural microgrid. It proposed a distributed, robust scheduling model for rural microgrids based on Wasserstein distance and on the cooperative interaction of source–grid–load–storage. Reference [5] proposed a smart building energy management system (BEMS), which was applied to a two-floor residential building, and the results show that the system could achieve energy self-sufficiency. This case shows that the model can achieve more economical and robust scheduling strategies compared to the stochastic optimization model based on a robust optimization model.

## 1.2. Construction of Micro-Energy Grid Optimization Scheduling Models

In terms of the optimization scheduling model construction of micro-energy grids, most of the research has carried out mathematical modeling on the infrastructure of micro-energy grids. Reference [6] established an efficient load-scheduling scheme by jointly considering an onsite photovoltaic (PV) system and an energy storage system (ESS), thereby reducing energy consumption costs. Reference [7] proposed a two-stage coordinated scheduling model to optimize multi-energy (electricity, heat, and cold) collaborative power supply in a microgrid. In order to realize the full utilization of renewable energy, [8] constructed a cooperative operation optimization model of a power distribution network with multiple microgrids and proposed an adaptive, dynamic, real-time optimization algorithm for the joint system based on pretraining and online deep learning technology. Reference [9] introduced plans for a solar, photovoltaic (PV) battery energy storage system (BESS) and a gas microturbine (MT) coupled with a micro-gas turbine and a power grid. It proposed a two-stage stochastic optimization model to help microgrid operators feasibly identify and optimize planning solutions. It also provided valuable guidance for energy infrastructure expansion from a comprehensive perspective. Reference [10] proposed a comprehensive optimization scheduling scheme for microgrids based on a predictive control model to eliminate the influence of uncertainty. The case simulation results demonstrate the effectiveness and feasibility of the proposed method. Reference [11] took the total cost (including investment, operation, maintenance, and fuel costs) and carbon dioxide emissions as the minimum optimization objective and proposed the optimal scale and operation strategy of a micro-energy power grid, using a multi-objective genetic algorithm (GA) to solve it. Reference [12] comprehensively considered the economy and reliability of micro-energy grids, constructing a two-layer optimization operation model aimed at obtaining an optimal operation cost and operation risk index and using mixed integer current programming combined with the IABC algorithm method to solve the model. Reference [13] considered the uncertainty of the output power of renewable energy. It proposed a data-driven, set-based Lupin optimization model considering the uncertainty of wind power and multi-demand responses. The calculation example shows that the model could improve the system's stability, as well as the performance and economy of the system's operation. In order to solve the problems of inaccurate, random, fluctuating, and intermittent load forecasting, Reference [14] proposed a three-stage, coordinated, and optimal dispatch strategy for the

CCHP microgrid. The result shows that the proposed strategy can reduce and eliminate the forecasting of renewable energy errors.

### 1.3. Operation Strategy of Micro-Energy Grids

Regarding micro-energy grid system operation strategies, [15] discussed the optimal operation of intermittent renewable energy in a micro-energy grid. Reference [16] considered the regular preventive maintenance measures of micro-energy grids and built a multi-objective optimization model to minimize cost and maximize system reliability. Reference [17] proposed an optimal operation model to minimize energy consumption and environmental cost. Reference [18] studied the demand response mechanism and divided demand response into price-based demand response and incentive-based demand response. Reference [19] integrated wind power, photovoltaics, gas turbines, and demand responses based on integrated energy into the operation of a multi-energy carrier system. Reference [20] adopted the method of robust optimization to deal with the prediction errors of renewable energy generation and market price. Reference [21] used a two-stage adaptive robust optimization model to solve the uncertainty of renewable energy output and load. Reference [22] proposed a point estimation method to describe the uncertainty of microgrids and used a robust optimization method to solve the model. Reference [23] proposed a two-stage Stackelberg game theory model, and the results show that the model had higher accuracy and better effect than the controlled autoregressive moving average (CARIMA) model. Reference [24] considered the optimal scheduling problem of microgrids with a high proportion of renewable energy systems (RES) and multiple energy storage systems (ESS); it introduced CVaR and proposed a two-stage optimal scheduling model considering economic and environmental protection. The results show that the model can not only reduce operating costs, but also provide decision support for decisionmakers through risk metrics.

Since a micro-energy grid integrates distributed renewable energy resources, the randomness and volatility of renewable energy output should be fully considered during the operation and management of the system, and it is necessary to develop a phased operation strategy according to the proximity of the time period during the operation process. At the same time, if we want to realize the autonomous operation and management of micro-energy grids as much as possible, it is crucial to consider the capacity allocation of flexible generation and storage devices in the day-ahead operation phase. However, few previous studies on micro-energy grids have considered this issue from the perspective of flexible capacity allocation. In addition, in the process of the clustering analysis of renewable energy output scenarios, the FCM-CCQ clustering method has been used not only to achieve scenario clustering, but also to evaluate different clustering schemes and to select the optimal clustering scheme.

With these motivations, this paper proposes a multistage optimization model for micro-energy grid operation, which contains day-ahead capacity allocation, intraday system scheduling, and real-time system scheduling. In the context of renewable energy scenario clustering analysis, we adopt a fuzzy c-mean-clustering comprehensive quality (FCM-CCQ) method to realize the reasonable evaluation and selection of the clustering results. At the same time, to further consider the impact of uncertainty factors on the system operation, the CVaR theory was introduced to transform the multistage stochastic optimization model and quantitatively evaluate the risk value brought on by uncertainty. Finally, a three-level coordinated scheduling optimization model considering CVaR multi-energy and multi-objective optimization is established, and an actual example is used to verify the effectiveness of the constructed model.

The main contributions of this paper include the following aspects:

(1)　Building a basic structural framework model of the micro-energy grid and explaining the mathematical model of the essential physical components;

(2) Building a three-level scheduling optimization model for micro-energy grids, which is divided into three stages: (a) day-ahead capacity configuration; (b) intraday system scheduling; (c) real-time system scheduling.

(3) Using the FCM-CCQ algorithm to describe the uncertainty of wind power and photovoltaics, the typical scenarios were obtained. The optimization calculation was carried out based on the typical scenarios.

(4) Based on CVaR theory, the risk value of the micro-energy grid system participating in spot market transactions was evaluated.

The rest of this paper is organized as follows: Section 2 describes the three-stage collaborative optimization operation framework for micro-energy grids and the connection mechanism between each stage; Section 3 describes the construction of the three-stage scheduling optimization model for the micro-energy grid discussed in this paper; Section 4 discusses and analyzes the results; and in Section 5, the conclusions are summarized, and future research directions and priorities are noted.

## 2. The Three-Stage Collaborative Optimization Operation Mechanism for a Micro-Energy Grid

In the model constructed for this paper, the assumptions made were that the micro-energy grid operator can achieve the deployment and use of all the operating equipment in the system, and the problem of multiple entities is not considered. The detailed modeling ideas are as follows.

### 2.1. The Three-Stage Collaborative Optimization Framework of a Micro-Energy Grid

The energy dispatch of the micro-energy grid belongs to the pre-dispatch; that is, the energy dispatch plan is established before the actual output of wind power and photovoltaics is known. Generally speaking, the day-ahead stage is mainly used to establish the reserve capacity allocation plan. The intraday stage is mainly used to establish the energy dispatch plan. The real-time stage is mainly used to correct the forecast deviation. Therefore, this paper designs a three-stage collaborative optimization structure for a micro-energy grid, as shown in Figure 1:

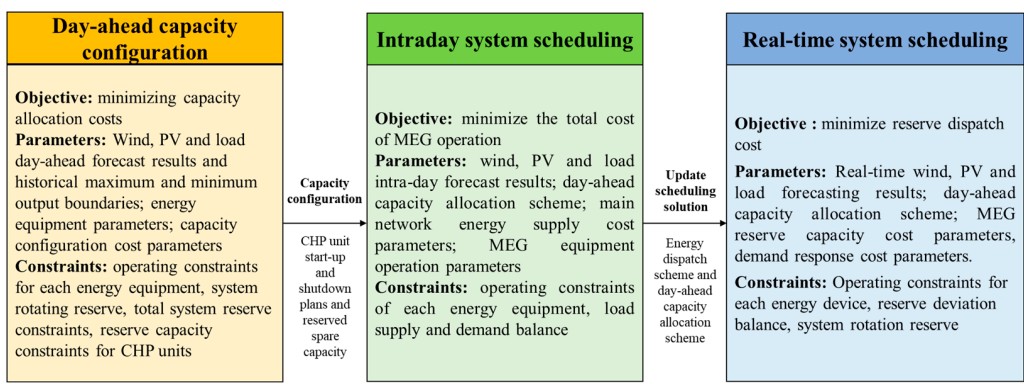

**Figure 1.** Three-stage collaborative optimization framework for a micro-energy grid.

Level 1: Day-ahead capacity configuration. According to the forecast value and error interval of wind power, photovoltaics, and load, each energy equipment in the micro-energy grid is configured. The purpose of the capacity allocation of the micro-energy grid is to realize the efficient supply of multiple loads, such as electricity, heat, cooling, and gas. This paper selects the minimum cost of spare capacity configuration as the optimization goal.

Level 2: Intraday system scheduling. Based on the capacity reconfiguration scheme of the micro-energy grid, the optimal supply of electricity, heat, cold, and gas in the micro-energy grid is considered according to the predicted value of wind power, photovoltaics, and load, and the goal of minimizing the operating cost of the micro-energy grid is realized.

In this paper, the optimization objective is to minimize the total dispatching cost of the micro-energy grid.

Level 3: Real-time system scheduling. Based on the energy scheduling results of the micro-energy grid, the deviation between the predicted value and the actual value of wind power and photovoltaics was analyzed. The reserve demand for electricity, heat, cold, and gas was determined. Finally, the micro-energy grid's optimal reserve capacity scheduling scheme is established. In this paper, the optimization objective is to minimize the real-time standby dispatching cost.

### 2.2. The Connection Mechanism of the Three Different Stage Models

The three-stage scheduling model of the micro-energy grid constructed above is an optimization problem in three different stages, but this does not mean that each stage is independent of the other two stages. There is a specific connection relationship between each stage. This section will explain in detail the connection mechanism of each optimization stage. The first-stage model is the day-ahead capacity configuration of the micro-energy grid. According to the simulation results of the first-stage model, it is substituted into the second-stage model and the intraday energy economic dispatch model, and is then substituted into the constraints as input parameters for the second-stage model. The calculation results of the models for the first and second stages are substituted into the constraints of the model of the third stage. According to the gas turbine start–stop plan cleared in the previous stage, the reserve capacity reserved for energy storage and demand response, as well as the production arrangement of each piece of equipment in the micro-energy grid, is adjusted based on the results of the clearing in the previous stage. Therefore, the third stage must be solved on the basis of the model results of the first and second stages and cannot be optimized separately.

## 3. Construction of a Three-Stage Scheduling Optimization Model for the Micro-Energy Grid

### 3.1. Uncertain Handling of Wind Power and Photovoltaic Output

This paper uses the FCM-CCQ clustering method to deal with the uncertainty of distributed wind power and photovoltaic output power [25]. The FCM-CCQ method is different from traditional clustering methods such as K-means, fuzzy c-means (FCM), etc. The traditional clustering method directly provides the number of cluster centers but cannot explain why this value is selected as the number of cluster centers. The FCM-CCQ method can solve this problem well. The FCM-CCQ method can be divided into two stages. The first stage uses the general clustering method to select the number of different cluster centers for clustering. The second stage evaluates the quality of each clustering result and, finally, guides decisionmakers to choose the best number of clustering centers, that is, the best clustering result. The specific solution steps are as follows:

Step 1: Similar to the conventional clustering method, first provide several clustering scenes and sample data for $n$ groups of wind speed scenes;

Step 2: Use the FCM method to perform clustering according to the specified number of scenes $a, 2a, 3a \cdots n - a$ and obtain corresponding multiple clustering schemes;

Step 3: Use the CCQ method to evaluate and score for each clustering scheme, calculate the two evaluation indicators of cluster density and cluster proximity, and then obtain the final result through weighted average. Different numbers of clusters correspond to different comprehensive quality scores;

Step 4: Draw the relationship curve between the comprehensive quality score and the number of clusters according to the calculation results. Find the inflection point of the curve to determine the best clustering scheme.

The specific solution process is shown in Figure 2. First, we select the initial minimum number of cluster centers as one, but this does not mean that our final clustering results are treated as one category; rather, only one of the clustering results is reserved. It should be noted here that the ultimate goal is to obtain a curve with the number of cluster centers

as the *x*-axis and the average comprehensive quality score as the *y*-axis. *b* is the weighted index, *t* is the index set, and *s* is the set of clustering results. Then, the historical data on wind power and photovoltaics are input, and the membership matrix is initialized to find the desired curve. The inflection point of the curve is the number of optimal clustering scenes.

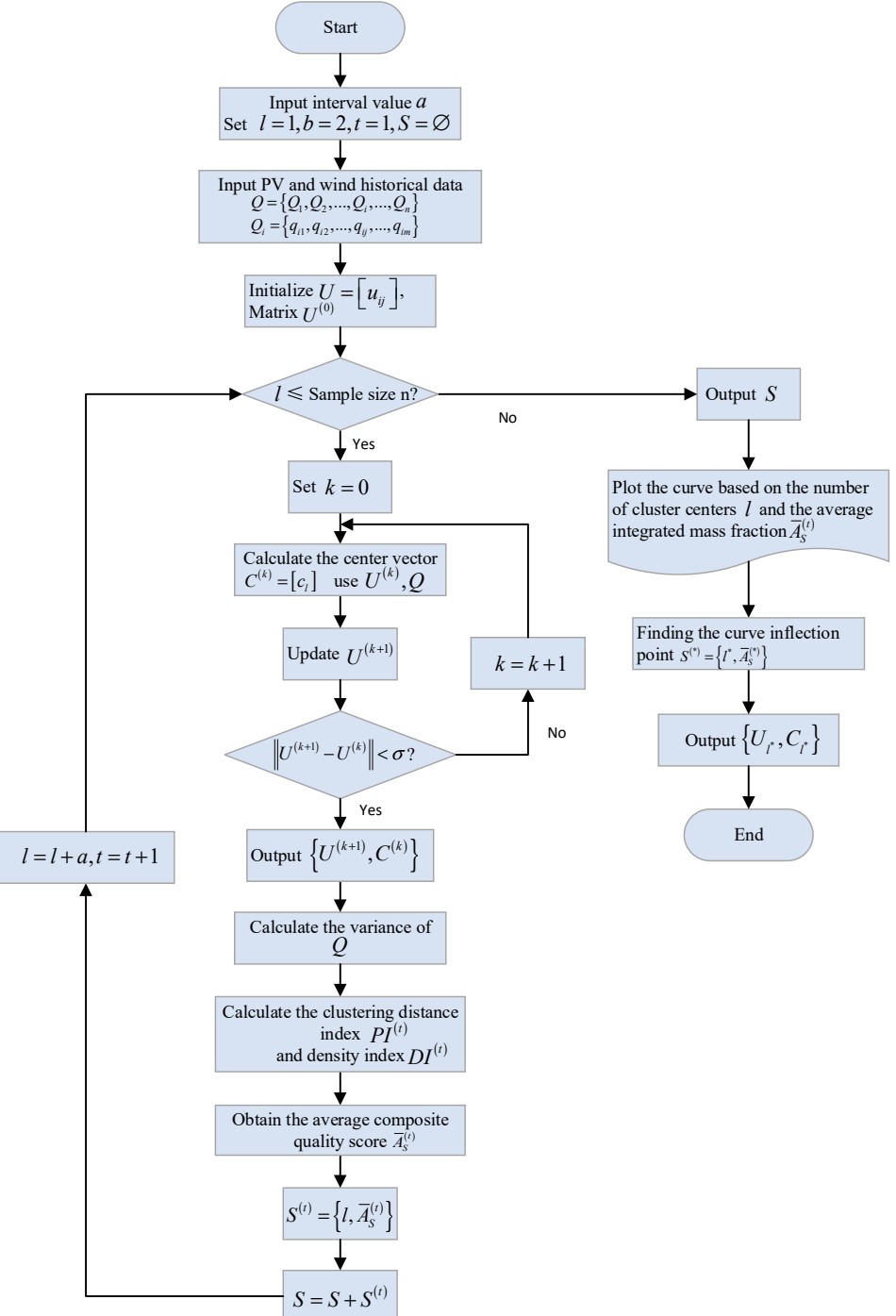

**Figure 2.** FCM-CCQ solution process.

## 3.2. The Construction of a Three-Stage Scheduling Optimization Model for Micro-Energy Grids
### 3.2.1. The First-Stage Model

The first stage is the optimization model for day-ahead capacity allocation. Because there are some renewable energy units in the micro-energy grid, such as wind power

and photovoltaics, in order to realize the priority consumption of renewable energy as much as possible and reduce the abandonment of wind and light resources, it is necessary to solve the randomness and intermittency of renewable energy output. Conventional thermal power units are generally used in traditional power grids to balance fluctuations in wind power output according to the peak-to-valley changes of load throughout the day. Sufficient thermal power reserve capacity is the key to ensuring the safe and stable operation of the power grid. However, there are many ways to stabilize the fluctuation of renewable energy output in the micro-energy grid. In this paper, a combination of gas turbines, power storage equipment, and demand response is used to stabilize the instability of renewable energy output.

(1) The Objective function In the day-ahead capacity allocation, since the electricity transaction process is not involved, it is only necessary to control the total cost of the micro-energy grid in the day-ahead capacity allocation stage. Therefore, the target aims to minimize the day-ahead capacity allocation cost. The objective function is shown in Equation (1):

$$minC_{co} = \sum_{t=1}^{T} \left[ \left( C_{GT,t} + C_{BESS,t} + C_{DR,t} \right) + \left( C_{spiil} \left( P_{W,t}^{spill} + P_{pv,t}^{spill} \right) \right) \right] \quad (1)$$

$$C_{GT,t} = C_{GT}^{startup} u_{GT,t}^{startup} + C_{GT}^{stop} u_{GT,t}^{stop} + C_{GT}^{RD} R_{GT,t}^{D} + C_{GT}^{RU} R_{GT,t}^{U} \quad (2)$$

$$C_{BESS,t} = C_{BESS,t}^{D} P_{dis,t} + C_{BESS,t}^{C} P_{chr,t} \quad (3)$$

$$C_{DR,t} = a + bP_{DR,t} \quad (4)$$

It can be seen from the objective function that the total cost ($C_{co}$) can be divided into three parts: gas turbine capacity allocation cost ($C_{GT,t}$), energy storage capacity allocation cost ($C_{BESS,t}$), and demand response capacity allocation cost ($C_{DR,t}$). Among them, the gas turbine capacity allocation cost includes the startup and shutdown cost of the unit, as well as the spare capacity cost.

(2) System constraints

(a) Gas turbine

$$P_{GT,min} \leq P_{GT,t} - R_{GT,t}^{D} \quad (5)$$

$$P_{GT,t} + R_{GT,t}^{U} \leq P_{GT,max} \quad (6)$$

$$R_{GT,t}^{D} \geq 0, R_{GT,t}^{U} \geq 0 \quad (7)$$

where $P_{GT,max}$ and $P_{GT,min}$ are, respectively, the maximum and minimum power of the gas turbine, while $R_{GT,t}^{U}$ and $R_{GT,t}^{D}$ are, respectively, the upper and lower reserves of the gas turbine. $P_{GT,t}$ represents the actual gas turbine power.

The functional relationship between the natural gas consumption and power generation of the gas turbine is shown in Equation (8):

$$F_{GT,t} = \frac{aP_{GT,t} + bu_{GT,t}}{\eta_{GT} \cdot LHV} \quad (8)$$

In the above formula, $a$ and $b$ are the gas-to-electricity conversion coefficients, is the power generation efficiency of the gas turbine, $LHV$ is the low calorific value of natural gas, and $u_{GT,t}$ is a 0–1 variable: take 0 to mean gas turbine shutdown; take 1 to mean startup. In addition, the power output constraints and ramp rate constraints of the gas turbine need to be considered. The mathematical expressions are shown in Equations (9) and (10):

$$u_{GT,t}P_{GT}^{min} \leq P_{GT,t} \leq u_{GT,t}P_{GT}^{max} \quad (9)$$

$$u_{GT,t}Ramp_{GT}^{down} \leq P_{GT,t} - P_{GT,t-1} \leq u_{GT,t}Ramp_{GT}^{up} \quad (10)$$

where $P_{GT}^{\min}$ and $P_{GT}^{\max}$, respectively, represent the upper and lower limits of the gas turbine output $Ramp_{GT}^{down}$ and $Ramp_{GT}^{up}$, respectively, represent the boundary value of the gas turbine ramping power between every two time periods. Considering that the gas turbine also produces heat energy while generating electricity, and the thermoelectric power relationship curve is generally a non-linear relationship, in order to facilitate the solution, this paper adopts the idea of piecewise linearization to linearize the thermoelectric power output relationship and convert it into a general mixed-integer programming problem [26]. The mathematical expressions are shown in Equations (11)–(14).

$$P_{GT,t} = u_{GT,t}M_{GT}^1 + \sum_{k=1}^{l} D_{GT,t}^k \tag{11}$$

$$u_{GT,t} = \sum_{k=1}^{l} z_{GT,t}^k \tag{12}$$

$$\sum_{m=k+1}^{l} z_{GT,t}^m \le \frac{D_{GT,t}^k}{M_{GT}^{k+1} - M_{GT}^k} \le \sum_{m=k}^{l} z_{GT,t}^m \tag{13}$$

$$H_{he} = u_{GT,t}N_{GT}^1 + \sum_{k=1}^{l} c_{GT}^k D_{GT,t}^k \tag{14}$$

where $M_{GT}^k$ is the end point electric power value of each segment after the piecewise linearization of the thermoelectric curve; $z_{GT,t}^m$ is a binary variable, indicating that the current operating state of the gas turbine is on the piecewise linear function of the $m$ segment; $c_{GT}^k$ is the slope of the linear function of the $k$ segment; and $H_{he}$ represents the heat produced by the gas turbine.

(b) Energy storage battery operation constraints

$$E_{t+1}^{BESS} = (1 - \eta_L^{BESS})E_t^{BESS} - \frac{P_{dis,t}\Delta t}{\eta_D} + \eta_C P_{chr,t}\Delta t, t = 0, 1, 2 \dots \tag{15}$$

$$E_{\min}^{BESS} \le E_t^{BESS} \le E_{\max}^{BESS} \tag{16}$$

$$0 \le P_{chr,t} \le u_{chr,t}^{BESS} P_{chr,\max} \tag{17}$$

$$0 \le P_{dis,t} \le u_{dis,t}^{BESS} P_{dis,\max} \tag{18}$$

$$u_{chr,t}^{BESS} + u_{dis,t}^{BESS} \le 1; u_{chr,t}^{BESS}, u_{dis,t}^{BESS} \in \{0,1\} \tag{19}$$

Equation (15) represents the state transition equation of the energy storage battery. $E_t^{BESS}$ represents the electrical energy stored by the energy storage battery in the time period $t$. $\eta_C, \eta_D$, and $\eta_L$ represent the charge and discharge efficiency of the energy storage battery and its self-discharge rate. $u_{chr,t}^{BESS}$ and $u_{dis,t}^{BESS}$ are the 0–1 state variables of the charging and discharging of the energy storage battery in the period $t$, respectively; 0 means that the behavior does not occur, and 1 means that the behavior occurs. $P_{chr,t}$ and $P_{dis,t}$ are the corresponding charging and discharging power of the energy storage battery in the period.

(c) Demand response capacity constraints

$$0 \le P_{DR,t} \le P_{DR,max} \tag{20}$$

where $P_{DR,t}$ represents the actual demand response reserve capacity signed with the power user during the $t$ period, and $P_{DR,max}$ represents the maximum corresponding capacity that the power user can bear.

(d)　　Abandon wind power and photovoltaic output power constraints

$$0 \leq P_{W,t}^{spill} \leq \widetilde{P}_{W,t} \tag{21}$$

$$0 \leq P_{pv,t}^{spill} \leq \widetilde{P}_{pv,t} \tag{22}$$

where $P_{pv,t}^{spill}$ and $P_{pv,t}^{spill}$ are the power generation of abandoned wind and photovoltaic, respectively, $\widetilde{P}_{W,t}$ and $\widetilde{P}_{pv,t}$ are the predicted values of wind power and photovoltaic output, respectively.

(e)　　System rotation reserve constraints: Reference [27] proposed that, in a distribution network with a high proportion of large-scale wind power, photovoltaics, and other highly volatile renewable energy units connected, the scheduling optimization model must consider the system spinning reserve constraints in the micro-energy grid. This constraint also needs to be considered. When there is a deviation in wind power and photovoltaic output, the reserved reserve capacity is enough to balance the deviation.

$$\begin{cases} R_{GT,t}^{U} + P_{dis,t} + P_{DR,t} \geq \widetilde{P}_{W,t} - P_{W,t}^{D} + \widetilde{P}_{pv,t} - P_{pv,t}^{D} \\ R_{GT,t}^{D} + P_{chr,t} \geq P_{W,t}^{U} - \widetilde{P}_{W,t} + P_{pv,t}^{U} - \widetilde{P}_{pv,t} \end{cases} \tag{23}$$

(f)　　System total reserve constraints: In the day-ahead stage, the total system reserve should be reserved for the intraday scheduling operation stage to ensure that the basic load requirements of cooling, heating, and electricity are met.

$$\eta_{ec} P_{ec,t} + Q_{D,t}^{IT} - Q_{C,t}^{IT} + \eta_{ac} Q_{ac,t} \geq (1 + \varphi_c) Q_{c,t} \tag{24}$$

$$\eta_{hc}(Q_{boiler,t} + Q_{rec,t} + Q_{dis,t} - Q_{chr,t} - Q_{ac,t}) \geq (1 + \varphi_h) Q_{h,t} \tag{25}$$

$$P_{pv,t} - P_{pv,t}^{spill} + P_{W,t} - P_{W,t}^{spill} + P_{GT,t} - P_{ec,t} \geq (1 + \varphi_e) P_{e,t} - P_{DR,t} \tag{26}$$

where $\varphi_c$, $\varphi_h$, and $\varphi_e$ are the reserve coefficients of the cooling, heating, and electric load of the micro-energy grid, respectively.

### 3.2.2. The Second-Stage Model

According to the solution result of the first-stage model, it should then be substituted into the second-stage model as the input parameter of the second-layer model.

(1)　The Objective function

The second-stage model considers two objectives, and the specific indicators can be reflected in the total cost of system operation and the $CO_2$ emissions of the micro-energy grid. However, due to the difference between the two target dimensions, this paper converts the $CO_2$ emissions into penalty costs to measure the economy of the micro-energy grid operation. These two goals can be expressed together, as shown in Equations (27)–(31):

$$min f_{ID} = \sum_{t=1}^{T} \left( C_{fuel,t} + C_{grid,t} + C_{rm,t} + C_{ce,t} \right) \tag{27}$$

$$C_{fuel} = fp(Fb + FGT) \tag{28}$$

$$C_{grid} = p_{M+} P_{M+} - p_{M-} P_{M-} \tag{29}$$

$$\begin{aligned} C_{rm} = rm_{gt} P_{GT} + rm_{ec} P_{ec} + rm_{rec} Q_{rec} + rm_b Q_b + rm_{bess}(P_{dis} + P_{chr}) \\ + rm_{tst}(Q_{dis} + Q_{chr}) + rm_{it}(Q_D^{IT} + Q_C^{IT}) + rm_s P_s + rm_w P_w \end{aligned} \tag{30}$$

$$C_{ce} = \sum_{t=1}^{T} pc(c_{gt} P_{GT} + c_{boiler} Q_b + c_{grid} P_{M+}) \tag{31}$$

where $C_{fuel}$, $C_{grid}$, $C_{rm}$, and $C_{ce}$, respectively, represent the fuel cost of the gas turbine, the power interaction cost between the micro-energy grid and the main grid, the operation and maintenance cost of the micro-energy grid, and the $CO_2$ emission penalty cost in the micro-energy grid. $Fb$ and $FGT$ represent the natural gas consumption of the gas turbine and gas boiler, respectively. $fp$ refers to the real-time purchase cost of natural gas. $P_{M+}$ and $P_{M-}$ decibels indicate the purchase and sale of electricity. $c_g$, $c_{boiler}$, and $c_{grid}$ represent the $CO_2$ emission intensity coefficients of gas turbines, gas boilers, and electricity purchased from the grid, respectively. $pc$ is the penalty coefficient. $rm_{gt}$, $rm_{ec}$, $rm_{rec}$, $rm_b$, $rm_{bess}$, $rm_{tst}$, $rm_{it}$, $rm_s$, and $rm_w$ represent the operation and maintenance costs per unit of electricity for gas turbines, electric refrigerators, waste heat boilers, gas boilers, batteries, heat storage tanks, ice storage machines, photovoltaics, and wind power, respectively.

(2) System constraints

$$u^*_{GT,t} P^{\min}_{GT} \leq P_{GT,t} - R^{D*}_{GT,t} \leq u^*_{GT,t} P^{\max}_{GT} \tag{32}$$

$$C_{ce} = \sum_{t=1}^{T} pc(c_{gt} P_{GT} + c_{boiler} Q_b + c_{grid} P_{M+}) \tag{33}$$

$$u^*_{GT,t}(Ramp^{down}_{GT} - R^{D*}_{GT,t}) \leq P_{GT,t} - P_{GT,t-1} \leq u_{GT,t}(Ramp^{up}_{GT} - R^{U*}_{GT,t}) \tag{34}$$

$$F_{GT,t} = \frac{a P_{GT,t} + b u_{GT,t}}{\eta_{GT} \cdot LHV} \tag{35}$$

$$P_{GT,t} = u^*_{GT,t} M^1_{GT} + \sum_{k=1}^{l} D^k_{GT,t} \tag{36}$$

$$H_{he} = u^*_{GT,t} N^1_{GT} + \sum_{k=1}^{l} c^k_{GT} D^k_{GT,t} \tag{37}$$

$$u^*_{GT,t} = \sum_{k=1}^{l} z^k_{GT,t} \tag{38}$$

$$\sum_{m=k+1}^{l} z^m_{GT,t} \leq \frac{D^k_{GT,t}}{M^{k+1}_{GT} - M^k_{GT}} \leq \sum_{m=k}^{l} z^m_{GT,t} \tag{39}$$

(3) Subsection

  (a) Gas turbine

  Compared to the expression of the gas turbine in the first-stage model, more variables are added here. Among them, $u^*_{GT,t}$, $R^{D*}_{GT,t}$, $R^{U*}_{GT,t}$ are the clearing results of the first stage, which are directly substituted here as boundary conditions.

  (b) Gas boiler

$$Q^{\min}_{boiler} \leq Q_{boiler,t} \leq Q^{\max}_{boiler} \tag{40}$$

$$F_{b,t} = \frac{Q_{boiler,t} \Delta t}{\eta_b \cdot LHV} \tag{41}$$

  where $Q_{boiler,t}$ is the thermal power output of the gas boiler at time $t$, $Q^{\max}_{boiler}$ and $Q^{\min}_{boiler}$ are the upper and lower limits of the output thermal power of the gas boiler, $F_{b,t}$ is the natural gas consumption of the gas boiler, $\eta_b$ is the energy conversion efficiency coefficient of the gas boiler.

  (c) Heat storage tank

  Similar to the energy storage battery, the heat storage tank also has a thermal energy state transfer equation, charge/discharge power constraints, and charge and discharge state constraints, as shown in Equations (42)–(46):



$$Q_{tst,t+1} = (1 - \eta_L^{TST})Q_{tst,t} - \frac{Q_{D,t}^{TST}\Delta t}{\eta_D^{TST}} + \eta_C^{TST}Q_{C,t}^{TST}\Delta t, t = 0, 1, 2\dots \quad (42)$$

$$Q_{tst}^{\min} \leq Q_{tst,t} \leq Q_{tst}^{\max} \quad (43)$$

$$0 \leq Q_{C,t}^{TST} \leq u_{chr,t}^{TST}Q_C^{TST,\max} \quad (44)$$

$$0 \leq Q_{D,t}^{TST} \leq u_{dis,t}^{TST}Q_D^{TST,\max} \quad (45)$$

$$u_{chr,t}^{TST} + u_{dis,t}^{TST} \leq 1 \quad (46)$$

(d) Ice Cooler
In the same way, the operating constraints of the ice-cold storage machine are as follows:

$$E_{t+1}^{ISS} = (1 - \eta_L^{ISS})E_t^{ISS} - \frac{P_{dis,t}^{ISS}\Delta t}{\eta_D} + \eta_C P_{chr,t}^{ISS}\Delta t, t = 0, 1, 2\dots \quad (47)$$

$$E_{\min}^{ISS} \leq E_t^{ISS} \leq E_{\max}^{ISS} \quad (48)$$

$$0 \leq P_{chr,t}^{ISS} \leq u_{chr,t}^{ISS}P_{chr,\max}^{ISS} \quad (49)$$

$$0 \leq P_{dis,t}^{ISS} \leq u_{dis,t}^{ISS}P_{dis,\max}^{ISS} \quad (50)$$

$$u_{chr,t}^{ISS} + u_{dis,t}^{ISS} \leq 1; u_{chr,t}^{ISS}, u_{dis,t}^{ISS} \in \{0,1\} \quad (51)$$

(e) Other energy conversion equipment
The micro-energy grid also includes three types of equipment: electric cooling, gas heating, and thermal cooling. Electric refrigeration equipment converts electrical energy into cold energy, and gas heating generates thermal energy by burning natural gas. These energy conversion devices have different energy efficiencies when switching energy types. For simplicity, this paper uses the form of a matrix to represent the energy conversion process, as shown in Equation (52):

$$\begin{bmatrix} Q_{ec,t} \\ Q_{rec,t} \\ Q_{h,t} \end{bmatrix} = \begin{bmatrix} P_{ec,t} & 0 & 0 \\ 0 & H_{he,t} & 0 \\ 0 & 0 & Q_{hc,t} \end{bmatrix} \begin{bmatrix} \eta_{ec} \\ \eta_{rec} \\ \eta_{hc} \end{bmatrix} \quad (52)$$

where $Q_{ec,t}$ is the cooling power generated by the electric refrigeration at time $t$. $Q_{rec,t}$ is the thermal power generated by the gas boiler at time $t$. $Q_{h,t}$ is the thermal power generated by the heating coil at time $t$. $P_{ec,t}$ and $H_{he,t}$, respectively, represent the electricity consumption of the electric refrigerator and the gas boiler at time $t$. $Q_{hc,t}$ represents the heat of the heating coil at time $t$. $\eta_{ec}$, $\eta_{rec}$, and $\eta_{hc}$ represent the energy conversion efficiencies of the three devices.

(f) Power balance constraints

$$\eta_{ec}P_{ec,t} + Q_{D,t}^{IT} - Q_{C,t}^{IT} = Q_{c,t} \quad (53)$$

$$Q_{rec,t} = \eta_{rec}H_{he} \quad (54)$$

$$Q_{h,t} = \eta_{hc}(Q_{boiler,t} + Q_{rec,t} + Q_{dis,t} - Q_{chr,t}) \quad (55)$$

$$P_{solar,t} + P_{wind,t} + P_{GT,t} - P_{ec,t} + P_{dis,t} - P_{chr,t} + P_{M-,t} - P_{M+,t} = P_{l,t} \quad (56)$$

where $P_{ec,t}$ represents the electric power consumed by the electric refrigerator at time $t$. $\eta_{ec}$ represents the conversion coefficient of the electric refrigeration machine to cold. $Q_{D,t}^{IT}$ and $Q_{C,t}^{IT}$ represent the cooling power and charging

power of the ice-cold storage machine at all times, respectively. $Q_{c,t}$, $Q_{h,t}$, and $P_{l,t}$ are the cooling load, heating load, and electrical load in the system at time $t$.

### 3.2.3. The Third-Stage Model

Based on the results of the first and second-stage models, the third stage is modeled with the goal of minimizing the total cost of real-time balancing.

(1) The Objective function

$$\min f_{RT} = \sum_{k=1}^{K} \rho_k \sum_{t=1}^{T} \left( \Delta C_{GT,k,t} + \Delta C_{BESS,k,t} + \Delta C_{DR,k,t} + \Delta C_{spill,k,t} + f_{ID} \right) \tag{57}$$

$$\Delta C_{GT,k,t} = p_{GT,t}^{+} \Delta_{GT,k,t}^{+} + p_{GT,t}^{-} \Delta_{GT,k,t}^{-} \tag{58}$$

$$\Delta C_{BESS,k,t} = p_{BESS,t}^{+} \Delta_{BESS,k,t}^{+} + p_{BESS,t}^{-} \Delta_{BESS,k,t}^{-} \tag{59}$$

$$\Delta C_{DR,k,t} = p_{DR,t} \Delta_{DR,k,t} \tag{60}$$

$$\Delta C_{spill,k,t} = C_{spill,t} \left( \Delta_{wind,k,t} + \Delta_{pv,k,t} \right) \tag{61}$$

It can be seen that the objective function is mainly divided into four parts: the cost of active power balance adjustment using gas turbines; the cost of active power balance adjustment using energy storage; the cost of using demand response management methods to reduce loads on the demand side; and penalty costs for abandoning wind and light. Where $\rho_k$ represents the probability of scenario occurrence, $f_{ID}$ is the total cost function of intraday economic dispatch.

(2) Stochastic optimization scheduling model considering CVaR

When a micro-energy grid is optimized for real-time balance scheduling, it is not only necessary to consider the issue of operating costs, but also to evaluate and analyze the risk of fluctuations in operating costs caused by uncertain factors. In this paper, CVaR theory is used to quantify the possible value-at-risk of the whole scheduling process. Therefore, based on the objective function (58), we add CVaR risk aversion, which is jointly optimized with the expected total cost through a linear combination. The mathematical expression of the objective function is shown in Equation (62):

$$\min_{X_t, \forall t} F = \lambda E(f_{RT}) + (1 - \lambda) CVaR_\alpha(f_{RT}) \tag{62}$$

where $\alpha$ is the confidence level, $\lambda$ is the weight coefficient, and then according to the transformation method of the literature [28], the Equation (62) is further transformed into the Equation (63):

$$\min_{X_t, \forall t} F = \lambda \sum_{k \in \Omega} \rho_k \left[ \sum_{t=1}^{T} \left( \Delta C_{GT,k,t} + \Delta C_{BESS,k,t} + \Delta C_{DR,k,t} + \Delta C_{spill,k,t} \right) \right] + (1 - \lambda) \left( \zeta + \frac{1}{\alpha} \sum_{k \in \Omega} \rho_k z_k \right) \tag{63}$$

$$s.t. \begin{cases} z_k \leq \sum_{t=1}^{T} \left( \Delta C_{GT,k,t} + \Delta C_{BESS,k,t} + \Delta C_{DR,k,t} + \Delta C_{spill,k,t} \right) - \zeta \\ z_k \leq 0 \end{cases} \tag{64}$$

where $\Omega$ represents the joint scenario set in the micro-energy grid and $k$ represents the kth scenario in it. $\rho_k$ represents the probability of occurrence of the *k*th scenario; $\zeta$, $z_k$ are intermediate parameters that have no practical significance.

In this paper, the system operating cost value is controlled between the expected operating cost and the CVaR value. When $\lambda = 0$, the micro-energy grid operator adopts a strategy of completely avoiding risks, fully considering the risks that may occur in the actual dispatching operation of the micro-energy grid, and avoids them. When $\lambda$ gradually increases, the strategy adopted by the micro-energy grid operator gradually tends to be risk-neutral. Finally, when $\lambda = 1$, it means that the scheduling optimization decision is

completely risk-neutral. At this time, only the expected total operating cost is considered, and the CVaR value is ignored.

(3) System Constraints

In the real-time balance stage, not only the constraints of intraday energy economic dispatch but also the constraints related to the real-time balance mechanism must be considered. The main constraints are shown in Equations (65)–(74):

$$0 \leq \Delta_{GT,k,t}^{+} \leq u_{GT,k,t}^{+} R_{GT,t}^{U*} \tag{65}$$

$$0 \leq \Delta_{GT,k,t}^{-} \leq u_{GT,k,t}^{-} R_{GT,t}^{D*} \tag{66}$$

$$u_{GT,k,t}^{+} + u_{GT,k,t}^{-} \leq 1 \tag{67}$$

$$0 \leq \Delta_{BESS,k,t}^{+} \leq u_{BESS,k,t}^{+} P_{dis,t}^{*} \tag{68}$$

$$0 \leq \Delta_{BESS,k,t}^{-} \leq u_{BESS,k,t}^{-} P_{chr,t}^{*} \tag{69}$$

$$u_{BESS,k,t}^{+} + u_{BESS,k,t}^{-} \leq 1 \tag{70}$$

$$E_{k,t+1}^{BESS} = \left(1 - \eta_{L}^{BESS}\right) E_{k,t}^{BESS} - \frac{\left(P_{chr,t}^{*} + \Delta_{BESS,k,t}^{+}\right)\Delta t}{\eta_{D}} + \eta_{C}\left(P_{chr,t}^{*} + \Delta_{BESS,t}^{-}\right)\Delta t \tag{71}$$

$$E_{min}^{BESS} \leq E_{k,t}^{BESS} \leq E_{max}^{BESS} \tag{72}$$

$$0 \leq \Delta_{DR,k,t} \leq P_{DR,t}^{*} \tag{73}$$

$$\Delta P_{W,k,t} + \Delta P_{pv,k,t} = \Delta_{GT,k,t}^{+} - \Delta_{GT,k,t}^{-} + \Delta_{BESS,k,t}^{+} - \Delta_{BESS,k,t}^{-} + \Delta_{DR,k,t} - \left(\Delta_{wind,k,t} + \Delta_{pv,k,t}\right) \tag{74}$$

Equations (65)–(67) represent the power adjustment constraints for the active power compensation of the gas turbine in the real-time phase, Equations (68)–(72) represent the power constraints for the energy storage to perform active power compensation in the real-time phase, and Equation (73) represents the demand response. With respect to the power constraint for active power compensation in the real-time stage, Equation (74) means that the deviation caused by wind power and photovoltaics in the real-time stage must ensure that there is enough capacity to balance.

*3.3. The Solution Method for the Three-Level Dispatch Optimization Model for the Micro-Energy Grid*

The overall solution to the three-level dispatch model of the micro-energy grid constructed in this paper can be roughly divided into four modules. The first module uses the scene-clustering method to describe the typical output scenarios of wind power and photovoltaics clustering into several reasonable typical scenarios. The second module is the solution of the day-ahead reserve capacity allocation model. The third module corresponds to the solution of the intraday economic dispatch operation model. The last module is based on the output results of the first three modules and is brought into the real-time balance stage dispatch model for a solution. A detailed flow chart is shown in Figure 3.

In terms of solving methods, the first module is the renewable energy output scenario clustering, which is solved by the FCM-CCQ algorithm, and the specific solution process is shown in Figure 2. Secondly, the models constructed in modules 2, 3, and 4 can be abstracted as mixed-integer linear programming problems, which can be solved by the branch-and-bound method, and since this method is a mature method in the field of operations research, the current stage can directly call upon CPLEX solver to scale up the solution. This method is not the highlight of this paper, so it will not be repeated.

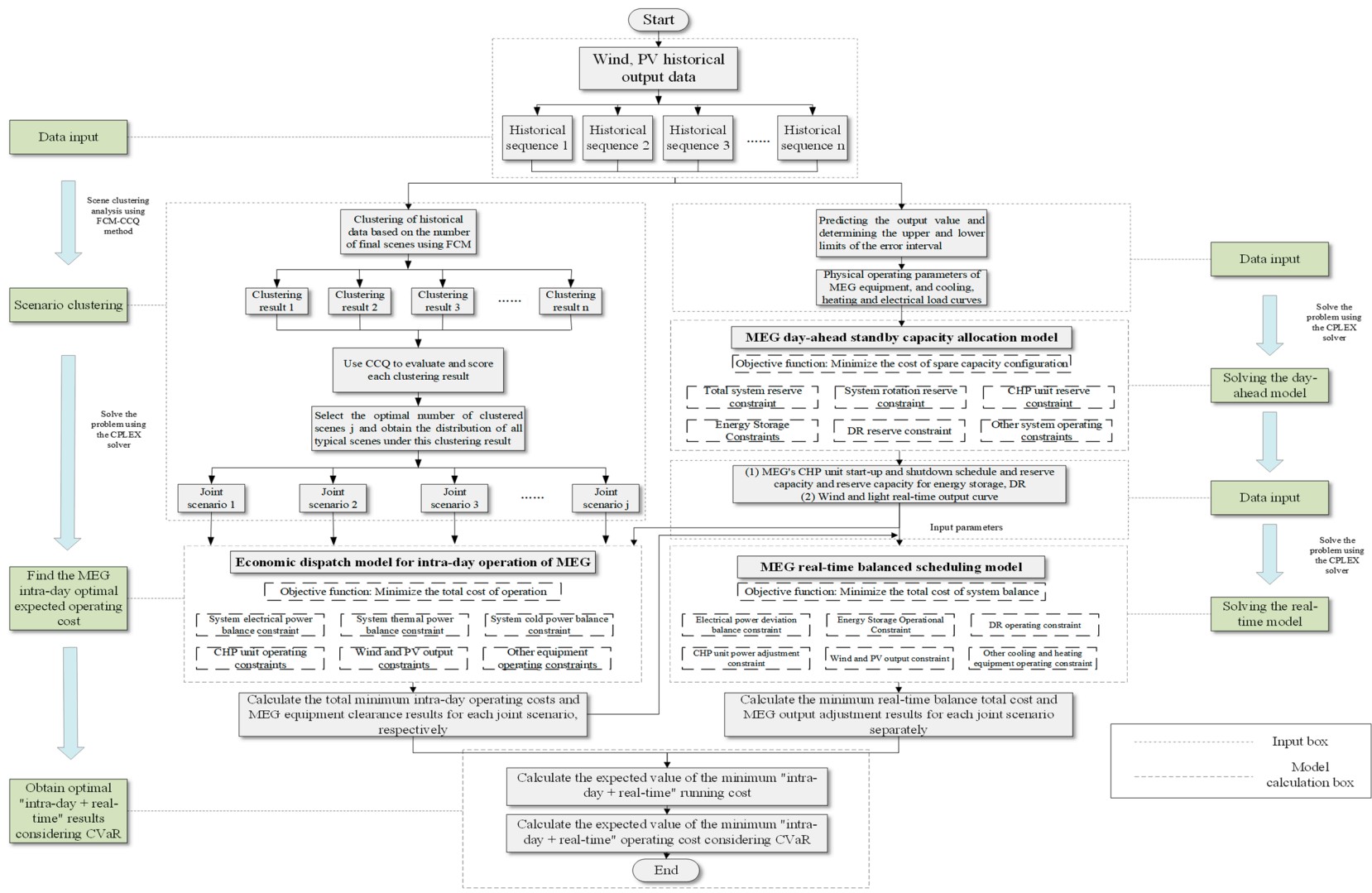

**Figure 3.** Solving the flow of the three-level scheduling optimization model for the MEG.

## 4. Example Analysis

### 4.1. Parameter Setting

In this paper, a small, low-carbon park was selected as an example analysis object. The park can be regarded as a micro-energy grid, and its structure is shown in Figure 4. The micro-energy grid includes a wind turbine with a rated power of 150 kW, a photovoltaic generator with a rated power of 150 kW, a cogeneration unit with a rated power of 250 kW, a 50-kW energy storage battery, a 500-kW gas boiler, a 160-kW heat storage tank, a 150-kW cold storage machine, a 300-kW absorption-type chiller, and a 100-kW electric chiller. Refer to [29] for the data on each piece of equipment; the cooling, heating, and electric load curves are shown in Figure 5. This example analysis was performed on a computer configured with an Intel(R) Xeon(R) E3-1230 v3 3.30 GHz and 8.00 GB of memory.

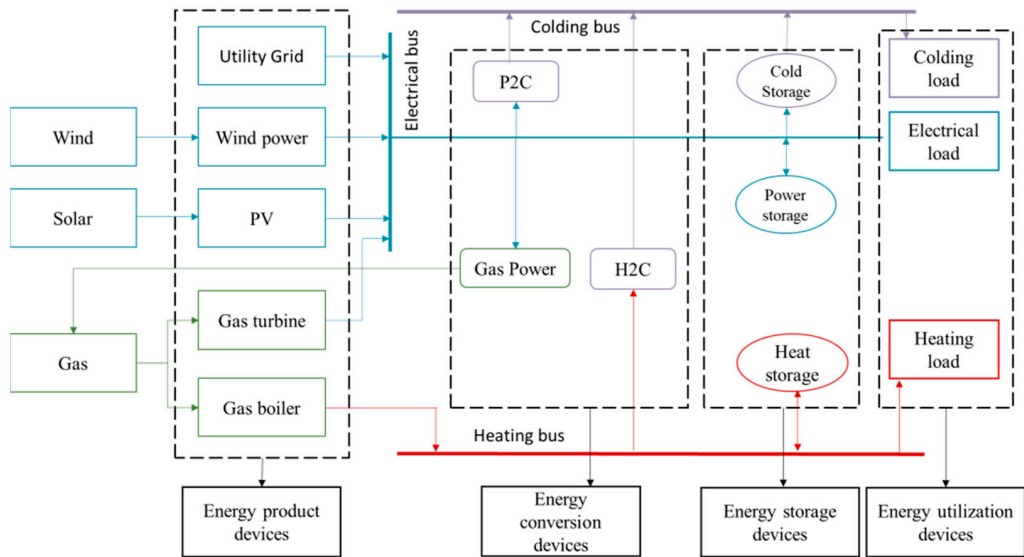

**Figure 4.** The structural framework of the MEG.

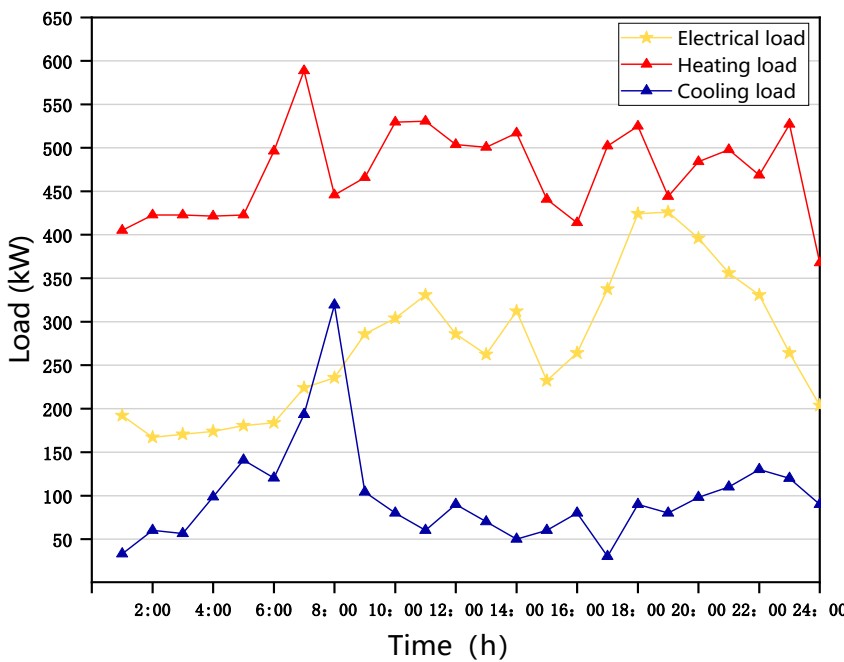

**Figure 5.** Curves of the cooling, heating, and electrical loads.

Since the real-time output scenarios of new energy need to be clustered in advance, this paper presents 50 groups of historical output data for wind power and photovoltaics at 24 h. The data come from a wind farm and a photovoltaic power station, as shown in Figures 6 and 7. The simulation of this example still uses MATLAB_2015b and calls for the CPLEX solver to solve the problem.

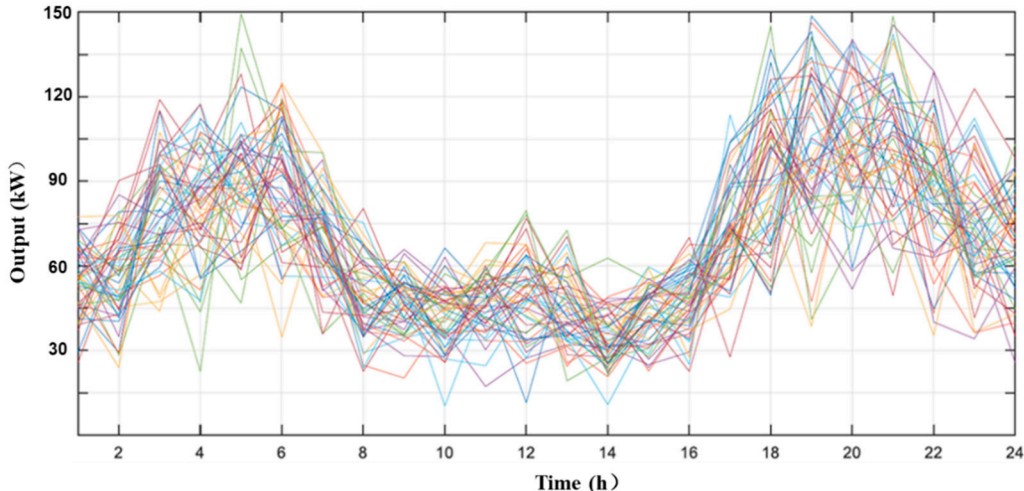

**Figure 6.** Fifty sets of wind power historical output data.

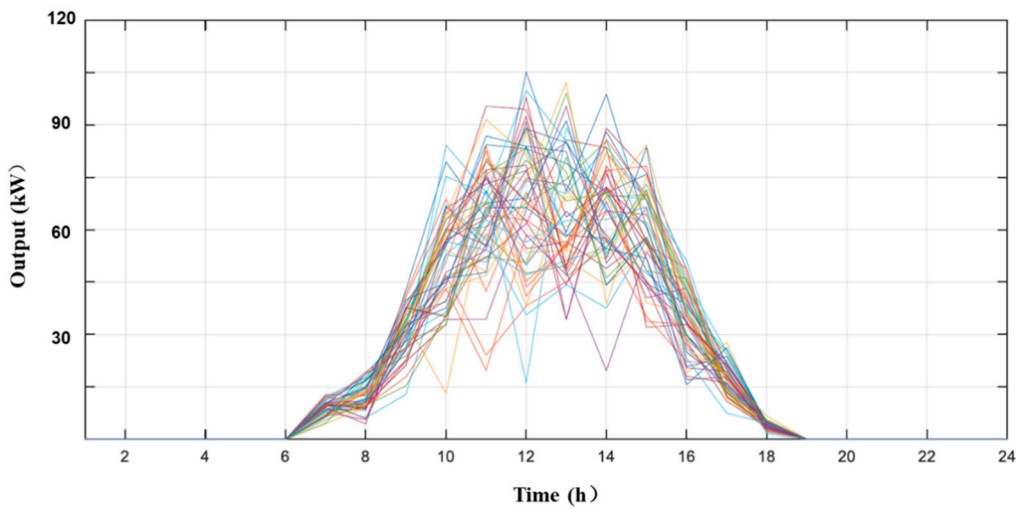

**Figure 7.** Fifty sets of PV historical output data.

### 4.2. Analysis of Results of Clearing the Day before

Based on the prediction of wind power, photovoltaics, and load data, as well as the analysis of the micro-energy grid capacity-optimized configuration scheme, the calculation results are shown in Figure 8. The solid black line in the figure represents wind power and photovoltaics at 24 h of possible maximum deviation; the sum of the solid gray lines represents wind power and photovoltaics under 24 h of the maximum possible deviation; and the sum of the bar chart represents the clearing result of the spare capacity of each spare piece of equipment. According to the results, the sum of deviations in the morning (around 6:00), noon (10:00–14:00), and evening (19:00–21:00) are larger than in other periods, which is consistent with the periods of large possible deviations in wind power and photovoltaics analyzed above, verifying the rationality of the results.

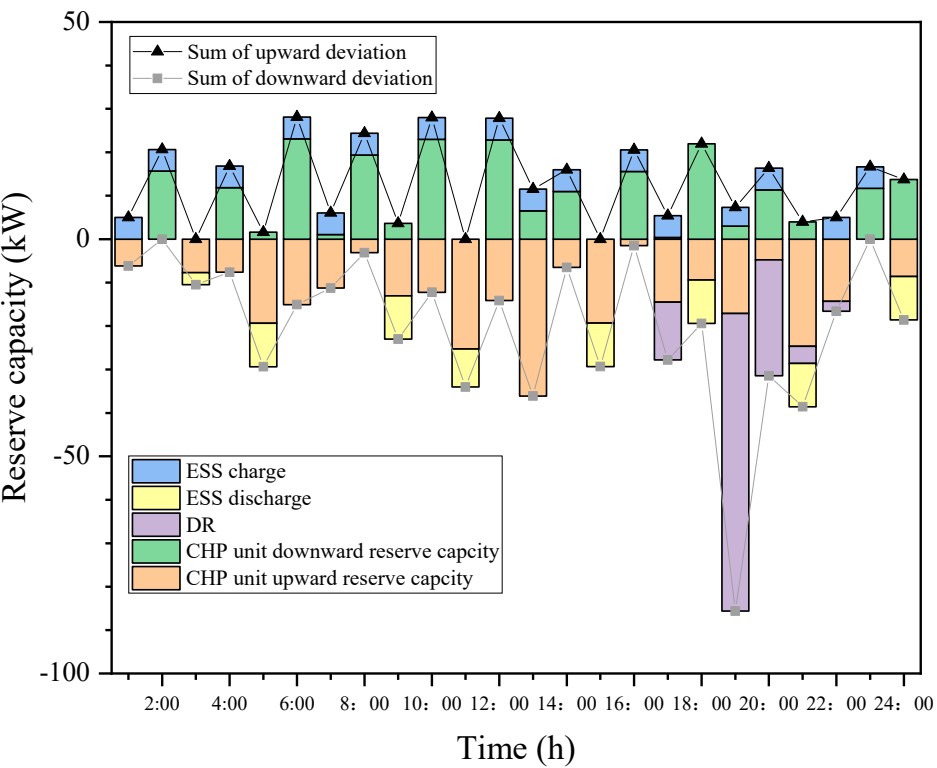

**Figure 8.** Reserve capacity clearance results.

Table 1 shows the cost of the gas turbine, demand response, and energy storage reserve capacities. From the perspective of cost structure, the cost of the reserve capacity of the gas turbine accounts for the highest proportion, exceeding 60%, and the leading role of the gas turbine in the entire reserve capacity allocation process can be obtained.

**Table 1.** Cost details of reserve capacity (unit: RMB).

| Gas Turbine Spare Capacity Cost | | Demand Response Spare Capacity Compensation Costs | Cost of Energy Storage Backup Capacity | | Total Reserve Capacity Cost |
|---|---|---|---|---|---|
| Up Spare Capacity Costs | Down Spare Capacity Costs | | Discharge Capacity Cost | Charging Capacity Cost | |
| 14,579.893 | 11,065.679 | 9181.017 | 2147.625 | 2379.639 | 39,353.854 |

*4.3. Analysis of Intraday Economic Dispatch Results*

As can be seen from Figure 9, since the marginal operating cost of wind power and photovoltaics is the smallest, the production plan is prioritized. However, the gas turbine is maintained at the minimum output level during the period of 0:00–4:00. At this time, it is necessary to consider the standby power. Capacity, that is, the minimum output, is equal to the sum of the spare capacity and the minimum technical output of the gas turbine at this moment. During the peak load period, from 18:00 to 22:00, the gas turbines are basically at full output, but there is still a power demand gap. Therefore, at this time, it is still necessary to purchase part of the electricity from the external power grid to meet the system demand. Figure 10 shows the scheduling results of thermal energy. It can be seen that the thermal storage tank displays thermal storage behavior during the periods of 8:00–9:00, 12:00, 15:00–16:00, and 19:00–20:00. To store the excess heat energy generated by the gas turbine, and heat is released from 22:00 to 24:00, which reduces the heat production of the gas boiler, realizes the decoupling of heat energy and electric energy to a certain extent, and avoids thermal energy loss, thus improving energy efficiency. Figure 11 shows the dispatching result of cold energy. It can be concluded that the ice-cold storage unit

purchases electric energy when the electricity price is low and converts it into cold energy; subsequently, cooling down at 8:00 a.m. reduces the power transmission pressure of the gas turbine during this period, realizes the time–space separation of supply and consumption, effectively avoids the high-power purchase price of peak load, saves power purchasing costs, and improves the operation of the micro-energy grid economy.

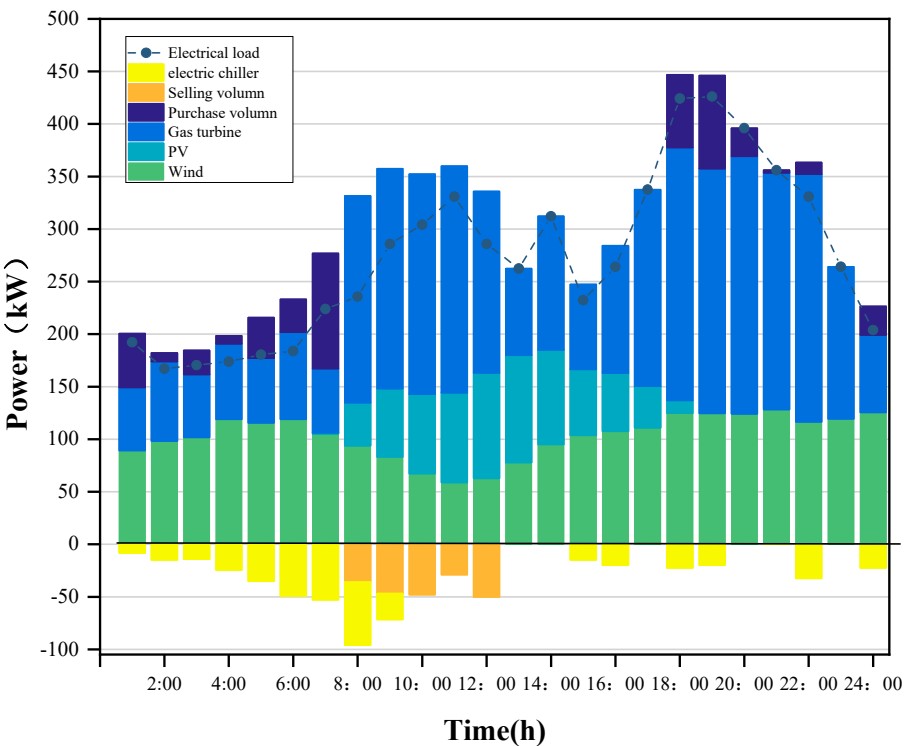

**Figure 9.** Power supply intraday dispatching results.

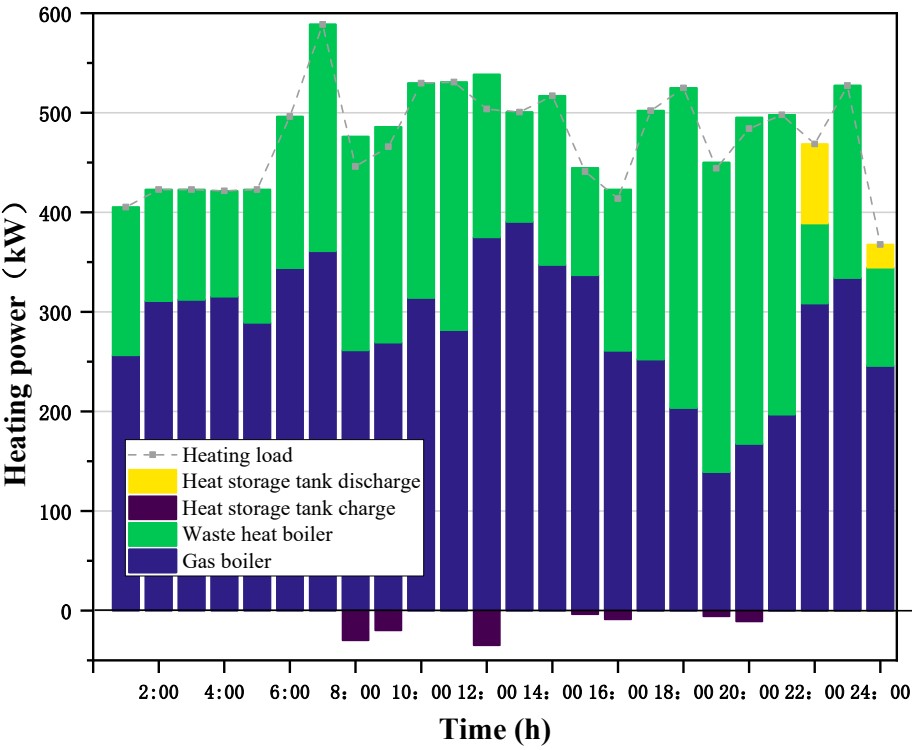

**Figure 10.** Heat supply intraday scheduling results.

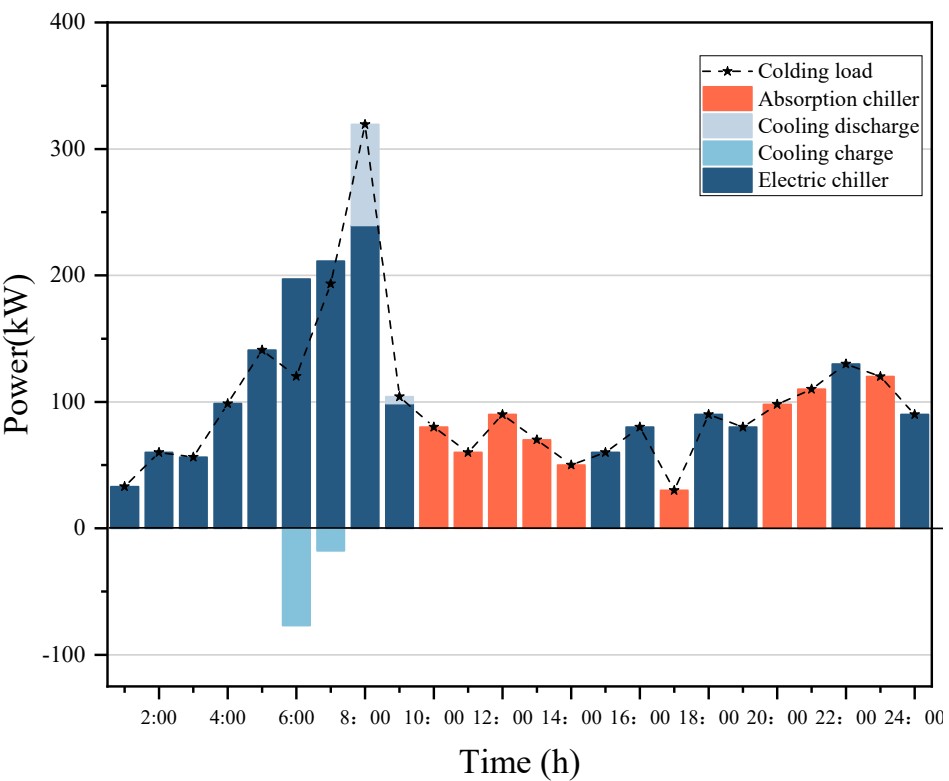

**Figure 11.** Cold energy supply intraday scheduling results.

### 4.4. Analysis of Clustering Results

The FCM-CCQ method is used to cluster historical output data on wind power and photovoltaics, and the marginal benefit curve of the cluster is obtained according to the clustering results, as shown in Figure 12.

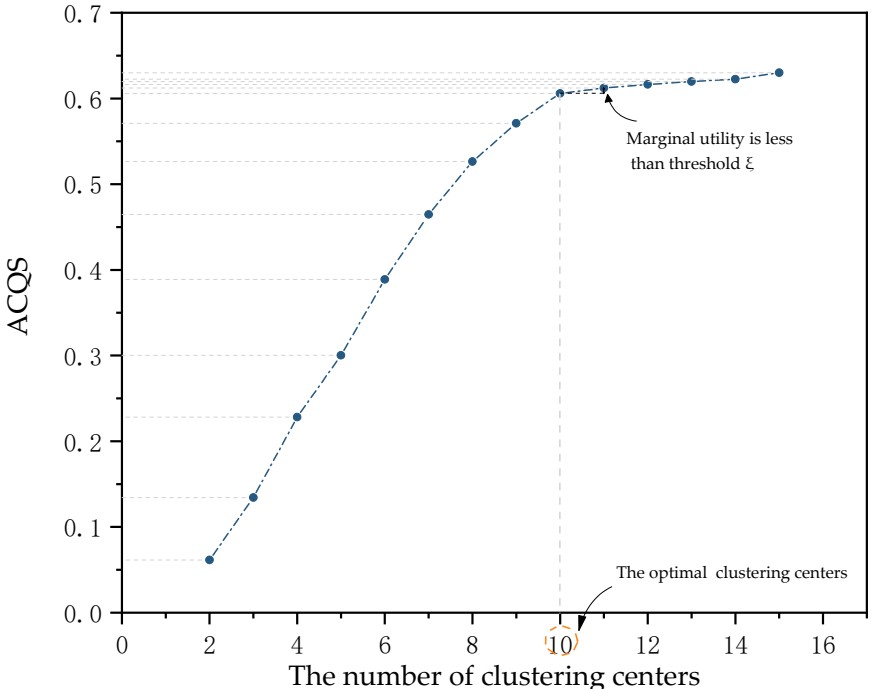

**Figure 12.** Clustering marginal benefit curve.

According to the curve in the figure, as the number of cluster centers increases, the average comprehensive quality score (ACQS) also increases gradually, but with a marginal decreasing effect. Further observation of the curve shows that when the marginal benefit exceeds a certain threshold, β, the corresponding boundary point is the optimal number of cluster centers. Then, according to the optimal number of cluster centers, the results and the probability of the typical clustering scenarios are obtained. The best clustering scenarios determined in this example are ten typical wind power and photovoltaic combined output scenarios, as shown in Figure 13:

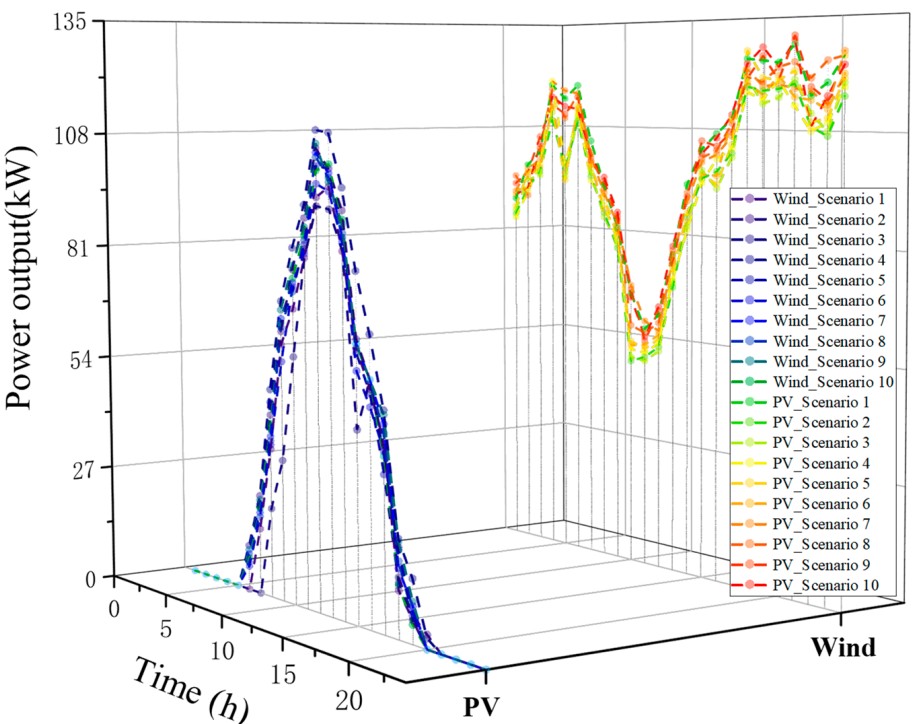

**Figure 13.** Ten typical output scenarios for wind turbines and photovoltaic units.

The probabilities of the above 10 typical scenarios are: 0.08, 0.12, 0.15, 0.05, 0.07, 0.13, 0.11, 0.09, 0.105, and 0.095. We needed to carry out real-time balanced optimal scheduling for the overall micro-energy grid under each typical scenario of the real-time output of the wind power and photovoltaics. However, since the scheduling optimization process is the same in each scenario, this paper analyzes the scheduling optimization results with one typical scenario out of ten wind power and photovoltaic output scenarios.

### 4.5. Analysis of Real-Time Stage-Clearing Results

According to the selected typical wind power and photovoltaic output scenarios, the curves of the actual and predicted output of wind power and photovoltaics in this scenario are obtained, as shown in Figure 14:

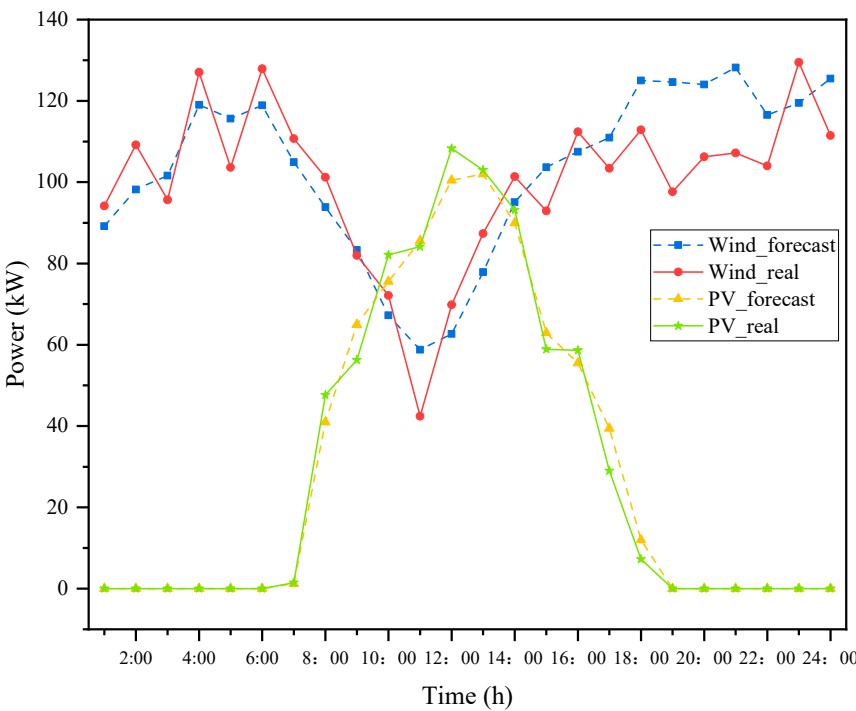

**Figure 14.** Curve comparison of the actual and predicted output of wind power and photovoltaic power.

Based on the actual output curve of the wind power and photovoltaics, the output results of each reserve capacity can be obtained, as shown in Figure 15:

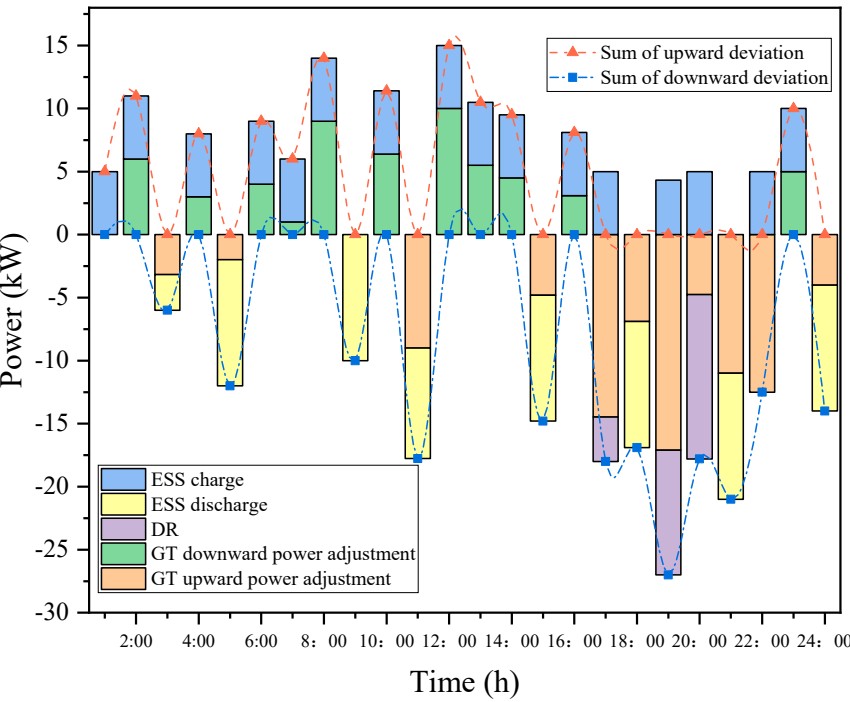

**Figure 15.** The result of the real-time phase standby capacity.

It can be seen from Figure 15 that the deviation between the actual output and the predicted value of the wind power and photovoltaics at different times can be compensated by the linkage adjustment mechanism of gas turbines, energy storage, and demand response. The energy storage still carries out orderly charging and discharging in accordance with the results and actual deviation of the day-ahead phase. The demand response is

also to reduce part of the electricity load according to the day-ahead phase and the agreed response period of the user. On the whole, the micro-energy grid uses its own energy equipment to completely balance the fluctuations of wind power and photovoltaic output, maintain the stability of the micro-energy grid operation, and play a safety role for the whole system.

As the output of the gas turbine in the micro-energy grid changes, the output results of other equipment in the real-time stage also change, as shown in Figures 16–18. Compared with the intraday economic dispatching results, it is not difficult to find that the output of gas turbines increases and decreases in different periods, which also means that the thermal power of gas turbines will increase and decrease correspondingly. In order to continue to meet the demand of thermal-load users, the output of the gas boiler will also change accordingly.

### 4.6. Sensitivity Analysis

(1) λ factor sensitivity analysis

Regarding the uncertainty of photovoltaic and wind power output, although it is described by scenario classification, the final operating cost is still only an expected value, and there are still some risks. Therefore, this paper uses CVaR risk control theory to reasonably avoid the economic risk of operating cost deviation. Since the subjective tendency coefficient, λ, in Formula (62) is set externally according to the degree of risk preference in the micro-energy grid, the value of λ will ultimately affect the decision result of the system. Figure 19 shows the Pareto boundary curve between the expected total operating cost and its CVaR for different values of λ. It can be seen that, as the value of λ increases, the decisionmaker tends to be risk-neutral; as the value of λ decreases, the decisionmaker tends to be risk-averse.

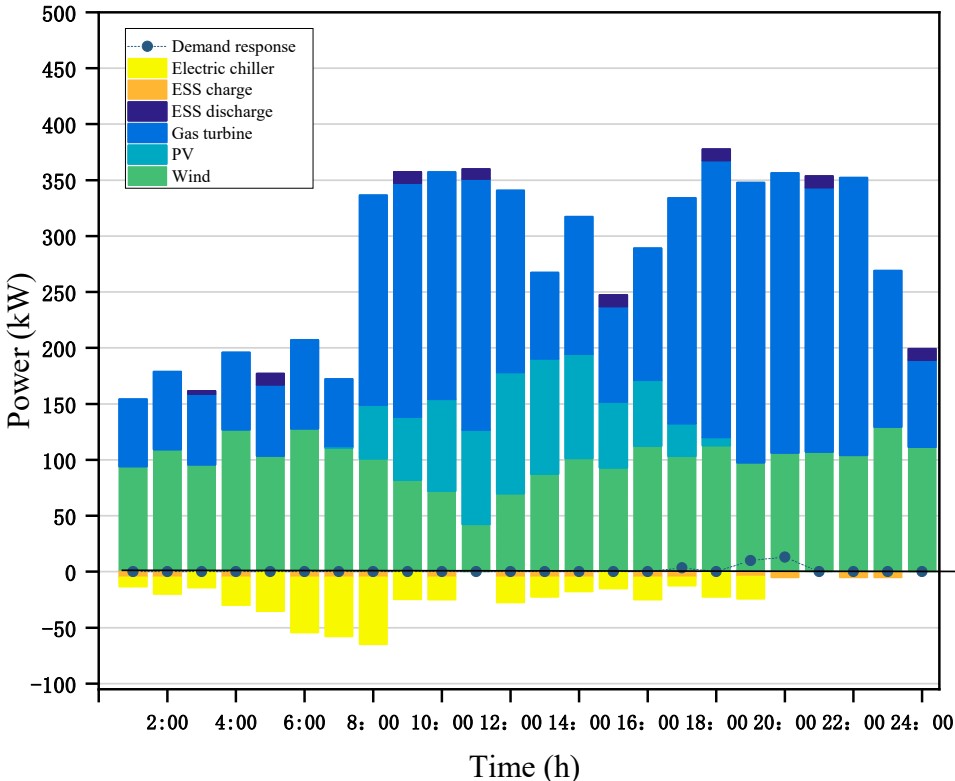

**Figure 16.** Real-time phase electrical energy clearance results.

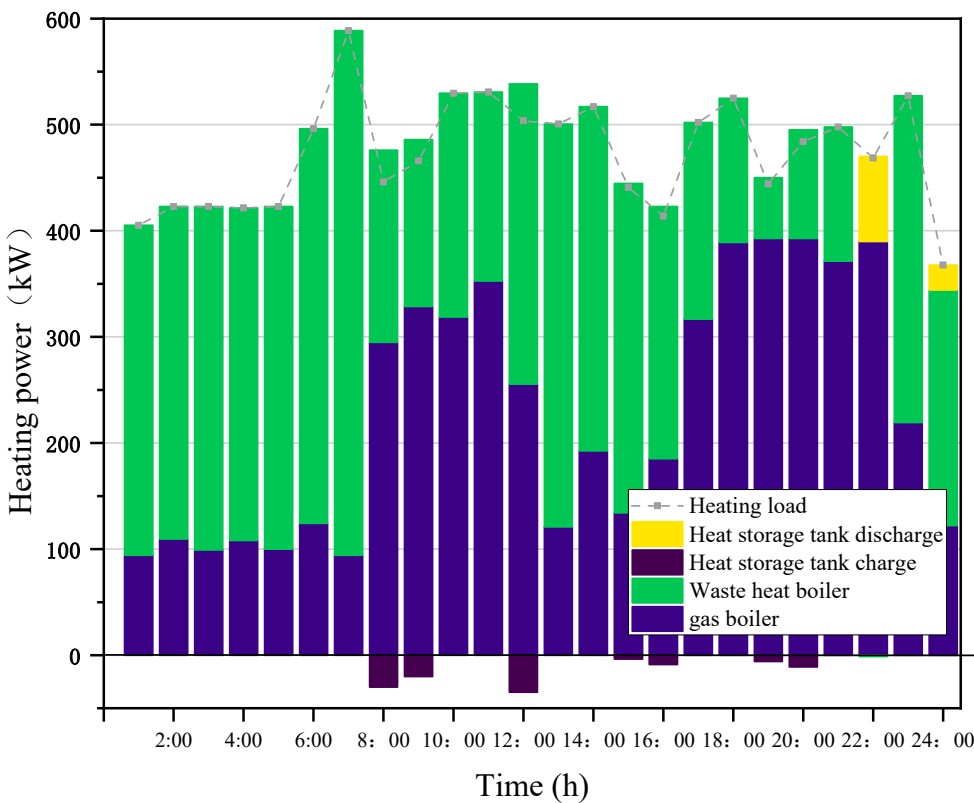

**Figure 17.** Real-time phase heating energy clearance results.

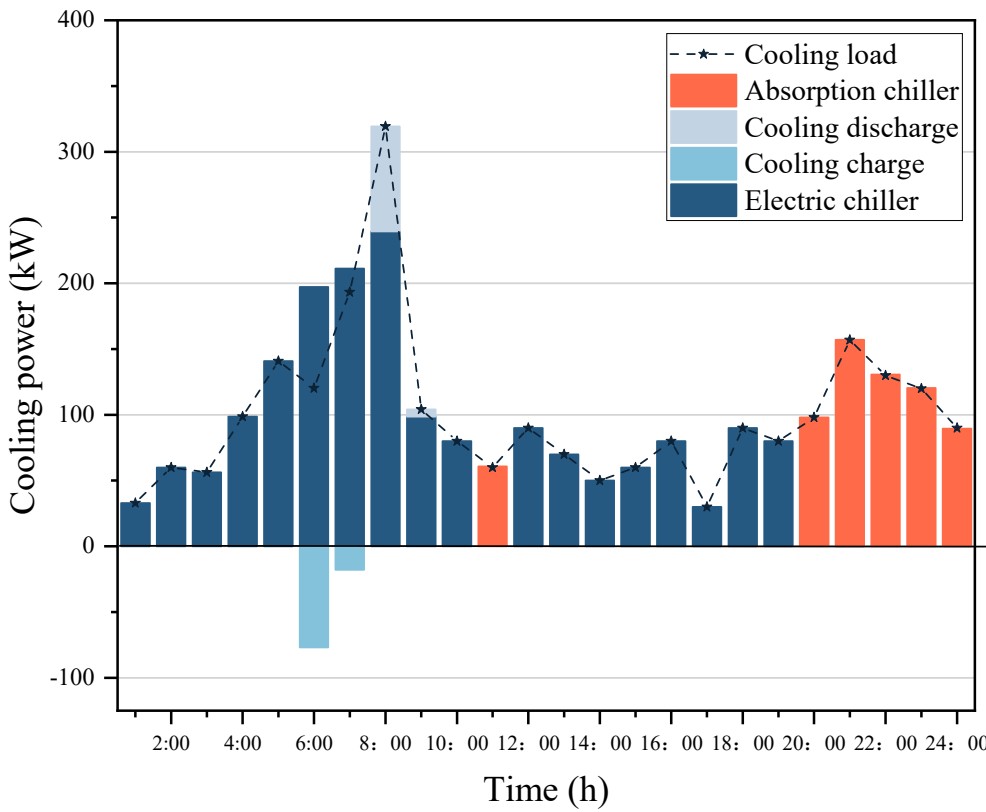

**Figure 18.** Real-time phase cooling energy clearance results.

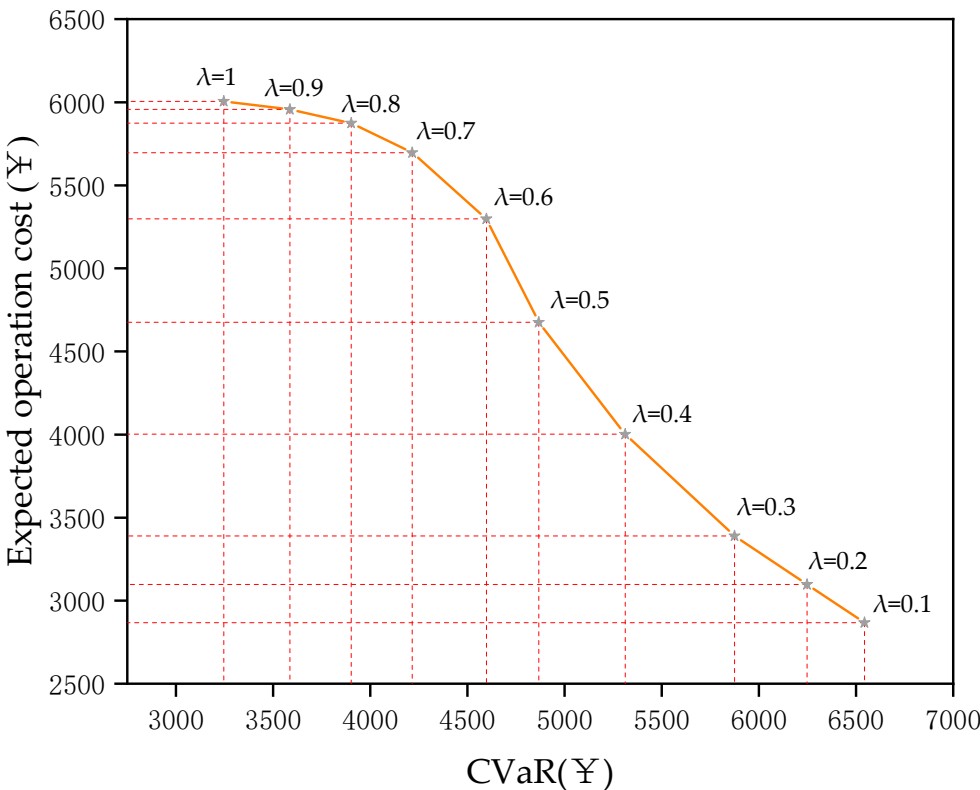

**Figure 19.** Pareto boundary curve of total cost and CVaR.

According to the curve analysis in the figure, we can see that the expected total cost of the real-time balance of the micro-energy grid will decrease with an increase in the λ value, but the CVaR value of its expected cost will decrease with an increase in the λ value. This is because when the value of λ is larger, it indicates that the micro-energy grid operators tend to choose the risk-neutral scheduling strategy, which can achieve the goal of minimizing the expected total cost but cannot resist the interference brought on by the uncertainty of the output power of renewable energy in the whole system. On the contrary, when the value of λ is smaller, micro-energy grid operators tend to adopt a more conservative scheduling strategy.

(2) Sensitivity analysis of carbon emission penalty coefficient

Bringing different penalty coefficient values into the model for further examination, $CO_2$ emissions under different penalty prices are obtained, as shown in Figure 20. As the government increases penalties for $CO_2$ emissions, micro-energy grids will gradually come to control $CO_2$ emissions; they will not always decline, but rather, gradually slow down and tend toward a stable value. Further analysis shows that $CO_2$ emissions in the micro-energy grid are closely related to the combustion of natural gas and the purchase of electricity from the main grid. It can be seen that, when the electricity and consumption purchased by external users reach a certain functional constraint, $CO_2$ emissions will not increase because the functional requirements set by the system will have been fully satisfied.

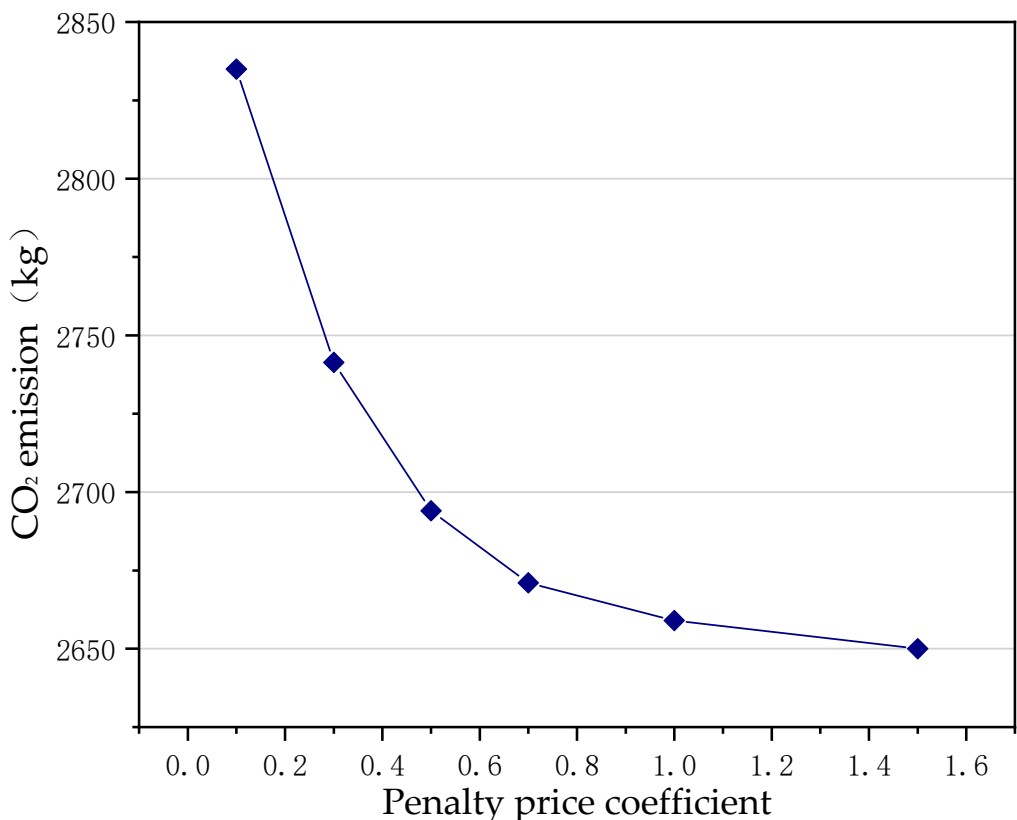

**Figure 20.** Relation curve between $CO_2$ emissions and the penalty price coefficient.

## 5. Conclusions

Micro-energy grids have great promotion value due to their relative independence, small scale, and flexibility. They can use different energy couplings to realize the coordinated supply of cold, heat, and electricity, as well as improve energy utilization efficiency through energy storage, energy conversion, demand response, and multi-energy complementation, as well as by reducing the peak-to-valley difference, thereby reducing the maximum capacity and reserve capacity of the energy supply side. This paper focuses on the multi-energy, collaborative, three-stage scheduling optimization operation of a micro-energy grid from the perspectives of operation benefit, operation cost, spare capacity, risk assessment, etc. The main research results and conclusions are as follows:

(1) A three-level scheduling optimization model framework for micro-energy grids is proposed. Firstly, the connection relationship between the models at all levels is described; secondly, the FCM-CCQ algorithm is used to obtain the optimal number of clustering centers, and then the typical scenarios and their probability distributions with respect to wind power and photovoltaic output are obtained; finally, based on the clustering results, in each typical scenario of combined wind power and photovoltaic output, a three-level dispatch optimization model of the micro-energy grid, with the minimum operating cost as the objective function, is constructed.

(2) Fully considering the risks brought on by uncertain factors to system operation, we use CVaR theory to transform the original model, realizing the visual display of risk value and guiding decisionmakers to adopt corresponding scheduling optimization strategies in a targeted manner to avoid possible risks;

(3) The practicability and effectiveness of the proposed model are verified by actual cases. The model can provide effective decision support for micro-energy grid operators under the condition of the electricity spot market.

The research work of this paper focuses on preconfiguring the capacity of flexible generation equipment and energy storage to cope with the uncertainty of renewable energy output in real-time, formulating an optimal strategy for micro-energy grid operation in

phases that can provide strategic suggestions and references for the autonomous operation and safety control of micro-energy grids. In addition, the clustering analysis method used in this paper, FCM-CCQ, has the dual function of clustering and evaluation, which is better than the traditional clustering algorithm and has a certain academic value. However, there are still some inadequate research areas in this paper, such as not considering the realistic problems that may be brought on by the operation of multiple entities in the micro-energy grid system. Subsequent research work can use the cooperative game theory to study the cooperative mechanism of the operation of multiple entities in the micro-energy grid.

**Author Contributions:** H.L. and Y.W. (Yongcheng Wang) supervised the proposed research work. S.N. conceptualized the proposed research work, collected the data, programmed and analyzed the methodology, and wrote the original draft. Y.W. (Yi Wang) and Y.C. helped in reviewing, writing, editing, and validating the research work. All authors have read and agreed to the published version of the manuscript.

**Funding:** This work is supported by the Foundation for Youth Scholars of Higher institution of Henan Province (2019GGJS179). This work is also supported by the Key Scientific and Technological Project of Henan Province (202102310311).

**Institutional Review Board Statement:** Not applicable.

**Informed Consent Statement:** Not applicable.

**Data Availability Statement:** Not applicable.

**Acknowledgments:** The authors would like to acknowledge the Key Scientific and Technological Project of Henan Province (202102310311). The authors would also like to acknowledge the Key Scientific Research Projects of Henan Higher Institutions (23A630035).

**Conflicts of Interest:** The authors declare no conflict of interest.

## Nomenclature

| | | | |
|---|---|---|---|
| CVaR | Conditional value at risk | $C_{fuel}$ | The fuel cost of the gas turbine |
| CCHP | Combined Cooling Heating and Power | $C_{grid}$ | The power interaction cost between the micro-energy grid and the main grid |
| PV | Photovoltaic | $C_{rm}$ | The operation and maintenance cost of the micro-energy grid |
| BESS | Battery energy storage system | $C_{ce}$ | The $CO_2$ emission penalty cost in the micro-energy grid |
| MT | Microturbine | $Fb$ | The natural gas consumption of gas boiler |
| GA | Genetic algorithm | $FGT$ | The natural gas consumption of gas turbine |
| CARIMA | Controlled autoregressive moving average | $fp$ | The real-time purchase cost price of natural gas |
| RES | Renewable energy systems | $P_{M+}$ | The purchase of electricity |
| ESS | Energy storage systems | $P_{M-}$ | The sale of electricity |
| FCM-CCQ | Fuzzy c-means-clustering comprehensive quality | $c_g$ | The $CO_2$ emission intensity coefficients of gas turbine |
| FCM | Fuzzy C-means | $c_{boiler}$ | The $CO_2$ emission intensity coefficients of gas boilers |
| ACQS | Average comprehensive quality score | $c_{grid}$ | The $CO_2$ emission intensity coefficients of electricity purchased from the grid |
| $C_{co}$ | The total cost | $pc$ | The penalty coefficient |
| $C_{GT,t}$ | Gas turbine capacity allocation cost | $rm_{gt}$ | The operation and maintenance costs per unit of electricity for gas turbines |

| | | | |
|---|---|---|---|
| $C_{BESS,t}$ | Energy storage capacity allocation cost | $rm_{ec}$ | The operation and maintenance costs per unit of electricity for electric refrigerators |
| $C_{DR,t}$ | Demand response capacity allocation cost | $rm_{rec}$ | The operation and maintenance costs per unit of electricity for waste heat boilers, |
| $P_{GT,max}$ | The maximum power of the gas turbine | $rm_b$ | The operation and maintenance costs per unit of electricity for gas boilers |
| $P_{GT,min}$ | The minimum power of the gas turbine | $rm_{bess}$ | The operation and maintenance costs per unit of electricity for batteries |
| $R_{GT,t}^U$ | The upper reserves of the gas turbine | $rm_{tst}$ | The operation and maintenance costs per unit of electricity for heat storage tanks |
| $R_{GT,t}^D$ | The lower reserves of the gas turbine | $rm_{it}$ | The operation and maintenance costs per unit of electricity for the ice storage machine |
| $P_{GT,t}$ | The actual gas turbine power | $rm_s$ | The operation and maintenance costs per unit of electricity for photovoltaic |
| $LHV$ | The low calorific value of natural gas | $rm_w$ | The operation and maintenance costs per unit of electricity for wind power |
| $u_{GT,t}$ | The 0–1 variable | $Q_{boiler,t}$ | The thermal power output of the gas boiler at time $t$ |
| $P_{GT}^{min}$ | The lower limits of the gas turbine output | $Q_{boiler}^{max}$ | The upper limits of the output thermal power of the gas boiler |
| $P_{GT}^{max}$ | The upper limits of the gas turbine output | $Q_{boiler}^{min}$ | The lower limits of the output thermal power of the gas boiler |
| $Ramp_{GT}^{down}$ | Lower boundary value of gas turbine ramping power | $F_{b,t}$ | The natural gas consumption of the gas boiler |
| $Ramp_{GT}^{up}$ | Upper boundary value of gas turbine ramping power | $\eta_b$ | The energy conversion efficiency coefficient of the gas boiler |
| $M_{GT}^k$ | The endpoint electric power value of each segment after the piecewise linearization of the thermoelectric curve | $Q_{ec,t}$ | The cooling power generated by the electric refrigeration at time $t$ |
| $z_{GT,t}^m$ | A binary variable | $Q_{rec,t}$ | The thermal power generated by the gas boiler at time $t$ |
| $c_{GT}^k$ | The slope of the linear function of the $k$ segment | $Q_{h,t}$ | the thermal power generated by the heating coil at time $t$ |
| $H_{he}$ | The heat produced by the gas turbine | $P_{ec,t}$ | The electricity consumption of the electric refrigerator at time $t$ |
| $E_t^{BESS}$ | The electrical energy stored by the energy storage battery in the time period $t$ | $H_{he,t}$ | The electricity consumption of the gas boiler at time $t$ |
| $\eta_C$ | The charge efficiency of the energy storage battery | $Q_{hc,t}$ | The heat of the heating coil at time $t$ |
| $\eta_D$ | The discharge efficiency of the energy storage battery | $P_{ec,t}$ | The electric power consumed by the electric refrigerator at time $t$ |
| $\eta_L$ | Its self-discharge rate | $\eta_{ec}$ | The conversion coefficient of the electric refrigeration machine to cold |
| $u_{chr,t}^{BESS}$ | The 0–1 state variables of the charging of the energy storage battery in the period $t$ | $Q_{D,t}^{IT}$ | The cooling power of the ice-cold storage machine at all times |

| $u_{dis,t}^{BESS}$ | The 0–1 state variables of the discharging of the energy storage battery in the period $t$ | $Q_{C,t}^{IT}$ | The charging power of the ice-cold storage machine at all times |
| --- | --- | --- | --- |
| $P_{chr,t}$ | The corresponding charging power of the energy storage battery in the period $t$ | $Q_{c,t}$ | The cooling load in the system at the moment $t$ |
| $P_{dis,t}$ | The corresponding discharging power of the energy storage battery in the period $t$ | $Q_{h,t}$ | The heating load in the system at the moment $t$ |
| $P_{DR,t}$ | The actual demand response reserve capacity signed with the power user during the $t$ period | $P_{l,t}$ | The electrical load in the system at the moment $t$ |
| $P_{DR,max}$ | The maximum corresponding capacity | $\rho_k$ | The probability of scenario occurrence |
| $P_{pv,t}^{spill}$ | The power generation of abandoned wind | $f_{ID}$ | The total cost function of intraday economic dispatch |
| $P_{pv,t}^{spill}$ | The power generation of abandoned photovoltaic | $\alpha$ | The confidence level |
| $\widetilde{P}_{W,t}$ | The predicted values of wind power output | $\lambda$ | The weight coefficient |
| $\widetilde{P}_{pv,t}$ | The predicted values of photovoltaic output | $\Omega$ | The joint scenario set in the micro-energy grid |
| $\varphi_c$ | The reserve coefficients of the cooling of the micro-energy grid | $k$ | The $k$-th scenario |
| $\varphi_h$ | The reserve coefficients of the heating of the micro-energy grid | $\rho_k$ | The probability of occurrence of the $k$-th scenario |
| $\varphi_e$ | The reserve coefficients of the electric load of the micro-energy grid | | |

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
