# Peer review of "Multistage Economic Scheduling Model of Micro-Energy Grids Considering Flexible Capacity Allocation"

_sustainability, doi:10.3390/su14159013_

Round 1

Reviewer 1 Report

The manuscript is written well and the concept of the work is clearly presented.  Some minor modifications can be done:'

(i) Figure 2 in the manuscript can be improved. 

(ii) A nomenclature can be added because a lot of abbreviated terms are used. 

(iii) Numerical results found in the work should be added in the abstract to highlight the impact of the work nicely. 

Author Response

Dear reviewer

Thank you for your decision and constructive comments on our manuscript. We have carefully considered the suggestion of Reviewer. Frankly, there are indeed a lot of deficiencies in our original manuscript, which hinder the accurate and effective display of our research results. The detailed suggestions for revision are very pertinent and helpful. Based on your comments, we have not only corrected the original manuscript, but also strengthened the scientific explanations for relevant problems. We sincerely hope the revisions can reach your understanding. Thank you very much for your following professional advice, hope our revisions can meet your requirements.

Figure 2 in the manuscript can be improved. 

Answer:

Thank you very much for your helpful suggestion, the Figure 2 has been improved. The clarity of the figure has been adjusted.

Question 2:

A nomenclature can be added because a lot of abbreviated terms are used.

Answer:

Thank you very much for your valuable comments, which we have revised in the manuscript, and Each abbreviated terms is annotated.Please see attachments.

Question 3:

Numerical results found in the work should be added in the abstract to highlight the impact of the work nicely.

Answer:

Thanks for this key comment. We have modified the abstract accordingly by adding the main conclusions of the numerical example to it. The additions are as follows.

With higher risk aversion factor, the more operational cost the system operator pays to avoid the risk. In addition, the higher the carbon penalty coefficient is, the overall carbon emission level of the micro-energy grid will decrease, but will gradually converge to a minimum level.

Reviewer 2 Report

Authors have done a great job in conducting their research and writing this manuscript. The manuscript is well written with a concreat logic. However, I have a concern on the overall novelty of the paper; the methodologies and the findings are quite general, without many insights. The authors have to defend their idea by clearly showing their novelty.

1. Please specify the importance of this research in Introduction section. Why is this research important? Do the authors' approaches have novelty?

2. Please provide more information on the selected example (the low-carbon park). Are the real-time output data from the wind power and the PV in the park? For how long are they measured?

3. The results and the main findings in this paper are too general. Please provde more context-specific findings and limitations of the authors' methodology taken in this research. Why are they important and how can they contribute to academia?

Author Response

Dear reviewer

Thank you for your decision and constructive comments on our manuscript. We have carefully considered the suggestion of Reviewer. Frankly, there are indeed a lot of deficiencies in our original manuscript, which hinder the accurate and effective display of our research results. The detailed suggestions for revision are very pertinent and helpful. Based on your comments, we have not only corrected the original manuscript, but also strengthened the scientific explanations for relevant problems. We sincerely hope the revisions can reach your understanding. Thank you very much for your following professional advice, hope our revisions can meet your requirements.

Question 1:

Please specify the importance of this research in Introduction section. Why is this research important? Do the authors' approaches have novelty?

Answer:

Thank you for this important comment. We indeed lacked a discussion of the importance and novelty of the paper in the introduction, and here we add a description of this section.

Since the micro energy grid integrates distributed renewable energy resources, the randomness and volatility of renewable energy output should be fully considered during the operation and management of the system, and it is necessary to develop a phased operation strategy according to the proximity of the time period during the operation process. At the same time, if we want to realize the autonomous operation and management of the micro energy grid as much as possible, it is crucial to consider the capacity allocation of flexible generation and storage devices in the day-ahead operation phase. However, few previous studies on micro energy grids have considered this issue from the perspective of flexible capacity allocation. In addition, in the process of clustering analysis of renewable energy output scenarios, the FCM-CCQ clustering method is used not only to achieve scenario clustering, but also to evaluate different clustering schemes and select the optimal clustering scheme.

Question 2:

Please provide more information on the selected example (the low-carbon park). Are the real-time output data from the wind power and the PV in the park? For how long are they measured?

Answer:

Thank you very much for your significant question. The numerical example we made is based on the data of the low carbon park, in which we collected 50 sets of output data for scenery, each set of data is the real-time output data of wind turbine and photovoltaic unit for one day, and the data recording point is one hour per day to record the output power data of wind turbine and photovoltaic for 24 hours.

Question 3:

The results and the main findings in this paper are too general. Please provide more context-specific findings and limitations of the authors' methodology taken in this research. Why are they important and how can they contribute to academia?

Answer:

Thank you for your comments and questions. The research work of this paper focuses on pre-configuring the capacity of flexible generation equipment and energy storage to cope with the uncertainty of renewable energy output in real time, and formulating the optimal strategy for micro energy grid operation in phases, which can provide strategic suggestions and references for autonomous operation and safety control of micro energy grid. In addition, the clustering analysis method used in this paper, FCM-CCQ, has the dual function of clustering and evaluation, which is better than the traditional clustering algorithm and has certain academic value. However, there are still some inadequate research areas in this paper, such as: not considering the realistic problems that may be brought by the operation of multiple entities in the micro energy grid system, and the subsequent research work can use the cooperative game theory to study the cooperative mechanism of the operation of multiple entities in the micro energy grid.

Reviewer 3 Report

This paper proposes a multi-stage economic scheduling model of multi-energy microgrid system. The constructions of the three stage optimization problems are described clear. The results show the effectiveness of the methods. There are some concerns as follows.

1)What kind of solution methodology is used to solve the optimization problem is not elaborated well.

2) For a multi-energy system, their operators (gas, heat, electricity, etc.) may be independent or have limited communication in the market. Can the proposed method be still effective?

Small problem:

3) FCM-CCQ is not explained when it occurs for the first time in Pg. 3.

Author Response

Dear reviewer

Thank you for your decision and constructive comments on our manuscript. We have carefully considered the suggestion of Reviewer. Frankly, there are indeed a lot of deficiencies in our original manuscript, which hinder the accurate and effective display of our research results. The detailed suggestions for revision are very pertinent and helpful. Based on your comments, we have not only corrected the original manuscript, but also strengthened the scientific explanations for relevant problems. We sincerely hope the revisions can reach your understanding. Thank you very much for your following professional advice, hope our revisions can meet your requirements.

Question 1:

What kind of solution methodology is used to solve the optimization problem is not elaborated well.

Answer:

Thank you for your helpful suggestions. According to Figure 3 in the paper, it can be seen that the solution process of the paper is divided into four major modules. The first module is the renewable energy output scenario clustering, which is solved by FCM-CCQ algorithm, and the specific solution process is shown in Fig. 2. Secondly, the models constructed in modules 2, 3 and 4 can be abstracted as mixed integer linear programming problems, which can be solved by branch-and-bound method, and since this method is a mature method in the field of operations research, the current stage can directly call CPLEX solver to scale up the solution. This method is not the highlight of this paper, so it will not be repeated in this paper.

Question 2:

For a multi-energy system, their operators (gas, heat, electricity, etc.) may be independent or have limited communication in the market. Can the proposed method be still effective?

Answer:

Thank you for your helpful question, which is extremely valuable, and we provide additional details on it here.

In the model constructed in this paper, the assumptions made are that the micro energy grid operator can achieve the deployment and use of all the operating equipment in the system, and the problem of multiple entities is not considered. This paper focuses on how the micro energy grid operator can achieve the economy of system operation through its own scheduling production arrangement when operating the system and participating in external market transactions. The multi-entity operation problem you mentioned is another issue worthy of subsequent in-depth study, which can be studied by adopting the framework of cooperative game or designing the system as a whole and finally sharing the revenue.

Thanks for your important comment.

Question 3:

FCM-CCQ is not explained when it occurs for the first time in Pg. 3.

Answer:

Thank you very much for your valuable comments. The Fuzzy C-Means-Clustering Integrated Quality method (FCM-CCQ), explained in Section 1.3, is a scene clustering method.

Round 2

Reviewer 2 Report

The importance of the work is now well explained as well as the limitations. The revised manuscript is much clearer than the original.